# TEDDY: TRIMMING EDGES WITH DEGREE-BASED DISCRIMINATION STRATEGY

**Hyunjin Seo[1]\*, Jihun Yun[1]\*, Eunho Yang[1,2]**
Korea Advanced Institute of Science and Technology (KAIST)[1], AITRICS[2]
{bella72,arcprime,eunhoy}@kaist.ac.kr

## ABSTRACT

Since the pioneering work on the lottery ticket hypothesis for graph neural networks (GNNs) was proposed in Chen et al. (2021), the study on finding graph lottery tickets (GLT) has become one of the pivotal focus in the GNN community, inspiring researchers to discover sparser GLT while achieving comparable performance to original dense networks. In parallel, the graph structure has gained substantial attention as a crucial factor in GNN training dynamics, also elucidated by several recent studies. Despite this, contemporary studies on GLT, in general, have not fully exploited inherent pathways in the graph structure and identified tickets in an iterative manner, which is time-consuming and inefficient. To address these limitations, we introduce TEDDY, a one-shot edge sparsification framework that leverages structural information by incorporating *edge-degree* statistics. Following the edge sparsification, we encourage the parameter sparsity during training via simple projected gradient descent on the $\ell_0$ ball. Given the target sparsity levels for both the graph structure and the model parameters, our TEDDY facilitates efficient and rapid realization of GLT within a *single* training. Remarkably, our experimental results demonstrate that TEDDY significantly surpasses conventional iterative approaches in generalization, even when conducting one-shot sparsification that solely utilizes graph structures, without taking feature information into account.

## 1 INTRODUCTION

Graph neural networks (GNNs) have emerged as a powerful tool for modeling graph-structured data and addressing diverse graph-based tasks, such as node classification (Kipf & Welling, 2016; Hamilton et al., 2017; Xu et al., 2018b; Wang et al., 2020; Park et al., 2021), link prediction (Zhang & Chen, 2018; Li et al., 2018; Yun et al., 2021b; Ahn & Kim, 2021; Zhu et al., 2021), and graph classification (Hamilton et al., 2017; Xu et al., 2018b; Lee et al., 2018; Sui et al., 2022; Hou et al., 2022). In conjunction with the notable performance achieved in GNNs, a substantial number of recent attempts have been made to handle large-scale real-world datasets. Owing to this, datasets and network architectures in graph-related tasks have progressively become more intricate, which incurs the notorious computational overhead both in training and inference.

In response to this challenge, GNN compression has emerged as one of the main research areas in GNN communities, and the Graph Lottery Ticket (GLT) hypothesis was articulated (Chen et al., 2021), serving as an extension of the conventional lottery ticket hypothesis (LTH, Frankle & Carbin (2019)) for GNN. Analogous to the conventional LTH, Chen et al. (2021) claimed that the GNNs possess a pair of core sub-dataset and sparse sub-network with admirable performance, referred to as GLT, which can be jointly identified from the original graph and the original dense model. In order to identify GLT, Chen et al. (2021) employ an iterative pruning as LTH where the edges/parameters are pruned progressively through multiple rounds until they arrive at the target sparsity level.

In parallel with advancements in GNN compression, a surge of recent studies has begun to underscore the increasing significance of graph structure over node features in GNN training dynamics. Notably, Tang & Liu (2023) have derived generalization error bounds for various GNN families. Specifically, they have discovered that the model generalization of GNNs predominantly depends on the graph

---

*Equal contribution.

structure rather than node features or model parameters. Furthermore, highlighted by the recent study (Sato, 2023), it has been both theoretically and empirically elucidated that GNNs can recover the hidden features even when trained on uninformative node features without the original node feature. Despite this, the preceding edge sparsification studies on identifying GLT have largely overlooked the paramount importance of graph structure.

Given the studies on GLT and the growing significance of graph structure, we introduce TEDDY, a novel edge sparsification framework that considers the structural information of the graph, aiming to maintain the primary message pathways. We begin by presenting the impact of degree information, which is observed in terms of both pruning performance and spectral stability. In alignment with our observation, TEDDY selectively discards graph edges based on scores designed by utilizing edge degrees. Our TEDDY integrates degree characteristics into the message-passing algorithm to carefully identify essential information pathways in multi-level consideration. Following this, our framework directly sparsifies parameters via projected gradient descent on $\ell_0$ ball. It should be noted that the sparsification of both edges and parameters in our TEDDY can be achieved within a single training, promoting enhanced efficiency compared to existing methods that require iterative procedures.

Our contributions are summarized as follows:

- We introduce TEDDY, a novel edge sparsification method that leverages structural information to preserve the integrity of principal information flow. In particular, our key observation for TEDDY lies in the importance of low-degree edges in the graph. We validate the significance of low-degree edges via comprehensive analysis.
- We encourage the parameter sparsity via projected gradient descent (PGD) onto $\ell_0$ ball. While conventional approaches identify the parameter tickets in an iterative fashion, only a single training process is required for our PGD strategy, which is much more computationally affordable.
- Our extensive experiments demonstrate the state-of-the-art performance of TEDDY over iterative GLT methods across diverse benchmark datasets and architectures. We note that TEDDY accomplishes one-shot edge pruning without considering node features, yet it asserts highly superior performance compared to the baselines that take node features into account.

## 2 RELATED WORK

Graph compression can be broadly categorized into two main approaches.

**Edge Sparsification** removes the edges in the graph with the number of nodes unchanged. In this line of work, in general, GNNs have differentiable masks for both graph and model parameter, and edges are pruned by the magnitudes of trained masks (Chen et al., 2021; You et al., 2022; Hui et al., 2023). As the first example, Chen et al. (2021) propose a unified framework for finding GLT by training $\ell_1$-regularized masks. To reach the target sparsity levels, they employ an iterative pruning which involves multiple pruning rounds gradually removing edges/parameters at each round. Due to the considerable time required for an iterative pruning, You et al. (2022) devise an algorithm to find GLT in an early pruning stage. More recent study (Hui et al., 2023) points out that the graph sparsity matters in GLT and proposes novel regularization based on Wasserstein distance between different classes, thereby searching better generalized GLT.

**Graph Coarsening**, in general, reduces the graph structure typically by grouping the existing nodes into super (or virtual) nodes and defining connections between them. Loukas & Vandergheynst (2018); Loukas (2019) propose randomized edge contraction, which provably ensures that the spectral properties of the coarsened graph and the original graph are close. Also, Chiang et al. (2019) cluster the entire graph into several small subgraphs and approximate the entire gradient via the gradient for the subgraphs, thereby increasing scability. Cai et al. (2021) introduce a novel framework that learns the connections between the super nodes via graph neural networks. Jin et al. (2022) propose graph condensation to reduce the number of nodes and to learn synthetic nodes and connections in a supervised manner. The recent study (Si et al., 2023) focus on graph compression in serving time, which constructs a small set of virtual nodes that summarize the entire training set with artificially modified node features.

The main purpose of this paper is to devise the *edge sparsification* algorithm that carefully considers the structural information of graph.

## 3 PRELIMINARIES

Before introducing TEDDY, we organize the problems of interest and necessary concepts in the paper.

### 3.1 PROBLEM SETUP

We consider an undirected graph $\mathcal{G} = (\mathcal{V}, \mathcal{E})$ which consists of $N = |\mathcal{V}|$ vertices and $M = |\mathcal{E}|$ edges. The vertex set $\mathcal{V}$ is characterized by a feature matrix $\boldsymbol{X} \in \mathbb{R}^{N \times F}$ with rows $\{\boldsymbol{x}_i\}_{i=1}^N$. In parallel, the edge set $\mathcal{E}$ is characterized by an adjacency matrix $\boldsymbol{A} \in \mathbb{R}^{N \times N}$ where $\boldsymbol{A}[i, j] = 1$ if an edge $e_{ij} = (i, j) \in \mathcal{E}$, and $\boldsymbol{A}[i, j] = 0$ otherwise. Given a graph $\mathcal{G} = (\mathcal{V}, \mathcal{E})$ and a feature matrix $\boldsymbol{X}$, a GNN $f_{\boldsymbol{\Theta}}$ learns the representation of each node $v$ by iteratively aggregating hidden representations of its adjacent nodes $u$ in the neighborhood set $\mathcal{N}_v$ in the previous layer, formulated as follows:

$$\boldsymbol{h}_v^{(l+1)} = \psi\big(\boldsymbol{h}_v^{(l)}, \ \phi(\{\boldsymbol{h}_u^{(l)}, \forall u \in \mathcal{N}_v\})\big) \tag{1}$$

where $\phi$ serves as an aggregation function, $\psi$ combines the previous representation of $v$ and aggregated neighbors. The initial representation is $\boldsymbol{h}_v^{(0)} = \boldsymbol{x}_v$ and we denote this multi-layered process by $f(\mathcal{G}, \boldsymbol{\Theta})$ or $f_{\boldsymbol{\Theta}}(\mathcal{G})$ where $\mathcal{G}$ is characterized by $\mathcal{G} = \{\boldsymbol{A}, \boldsymbol{X}\}$. Our focus is a semi-supervised node classification task, and the objective function $\mathcal{L}$ is defined as the cross-entropy loss between the prediction $\boldsymbol{P} = \text{softmax}(\boldsymbol{Z}) = \text{softmax}\big(f(\mathcal{G}, \boldsymbol{\Theta})\big) \in \mathbb{R}^{N \times C}$ and the label matrix $\boldsymbol{Y} \in \mathbb{R}^{N \times C}$:

$$\mathcal{L}(\mathcal{G}, \boldsymbol{\Theta}) = -\frac{1}{N} \sum_{i \in \mathcal{V}}^{N} \sum_{k=1}^{C} \boldsymbol{Y}_{ik} \log \boldsymbol{P}_{ik}, \tag{2}$$

where $\boldsymbol{Z}$ denotes the representation from the GNN, and $C$ represents the total number of classes.

### 3.2 GRAPH LOTTERY TICKET (GLT)

In recent study (Chen et al., 2021), the conventional lottery ticket hypothesis (LTH, Frankle & Carbin (2019)) was extended to graph representation learning. Analogous to conventional LTH, which asserts that one can discover sparse subnetworks with comparable performance to dense networks, Chen et al. (2021) claim that both core sub-dataset and sparse sub-networks can be simultaneously identified, called graph lottery ticket (GLT). Formally, graph lottery ticket can be characterized as follows.

**Definition 1** (Graph Lottery Tickets (GLT), Chen et al. (2021)). *Given a graph $\mathcal{G} = \{\boldsymbol{A}, \boldsymbol{X}\}$ and GNN $f_{\boldsymbol{\Theta}}(\cdot)$ parametrzied by $\boldsymbol{\Theta} \in \mathbb{R}^d$, let $\boldsymbol{m}_g \in \{0, 1\}^{N \times N}$ and $\boldsymbol{m}_\theta \in \{0, 1\}^d$ be the binary masks for $\boldsymbol{A}$ and $\boldsymbol{\Theta}$ respectively. Let $\boldsymbol{\Theta}' = \boldsymbol{m}_\theta \odot \boldsymbol{\Theta}$ be a sparse parameter and $\mathcal{G}' = \{\boldsymbol{A}', \boldsymbol{X}\}$ be a sparse subgraph where $\boldsymbol{A}' = \boldsymbol{m}_g \odot \boldsymbol{A}$ with $\|\boldsymbol{A}'\|_0 < M$ and $\|\boldsymbol{\Theta}'\|_0 < d$. If a subnetwork $f_{\boldsymbol{\Theta}'}(\cdot)$ trained on a subgraph $\boldsymbol{A}'$ has a comparable or superior performance to the original GNN $f_{\boldsymbol{\Theta}}(\cdot)$ trained on the entire graph $\mathcal{G}$, then we call $\boldsymbol{A}'$ and $f_{\boldsymbol{\Theta}'}$ as graph lottery ticket (GLT).*

After the proposal of GLT, numerous studies (Chen et al., 2021; You et al., 2022; Hui et al., 2023; Wang et al., 2023) have been carried out to identify more sparse masks $\boldsymbol{m}_g$ and $\boldsymbol{m}_\theta$ with better performance. The most studies identify GLT in an iterative manner, in which the edges/parameters are gradually removed at each round and GLT with desired sparsity levels is realized after multiple rounds. However, a majority of studies falls short in contemplating the graph structure information.

## 4 MOTIVATION: LOW-DEGREE EDGES ARE IMPORTANT

**Empirical observations.** We first highlight a critical graph structural property that profoundly impacts the performance of the graph sparsification. Toward this, we compare the performances of GAT (Veličković et al., 2017) with two simple edge pruning strategies: pruning graph edges by (1) the highest edge degree and (2) the lowest edge degree. Throughout our observations, we define the edge degree of $e = (i, j)$ as $(|\mathcal{N}(i)| + |\mathcal{N}(j)|)/2$ where $\mathcal{N}(k)$ denotes the set of neighboring nodes connected to the node $k$. As illustrated in Figure 1, pruning low-degree edges significantly degrades the performance compared to the pruning of high-degree edges. In particular, the performance gap gradually increases approaching to high sparsity regime, especially in Citeseer and Pubmed datasets. Similar observations can be found for other standard GNN architectures such as GCN (Kipf & Welling, 2016) and GIN (Xu et al., 2018a), and they are provided in in Appendix A.1.

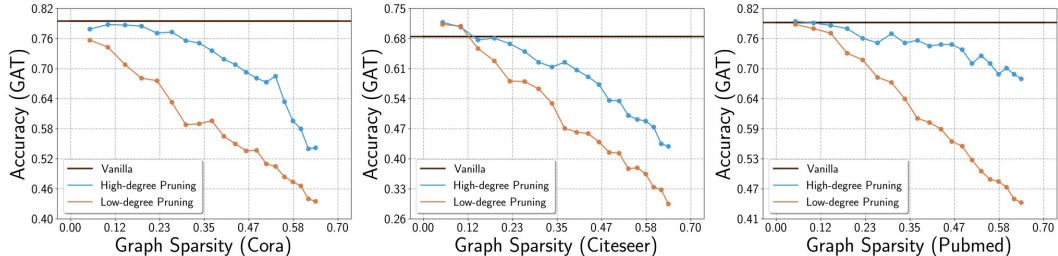

Figure 1: Empirical observations for the importance of edge degrees.

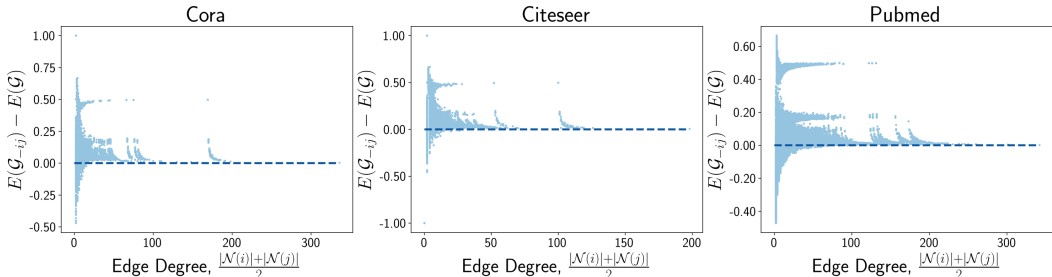

Figure 2: The changes of Laplacian energy for single edge removal on Cora/Citeseer/Pubmed datasets.

**Normalized graph Laplacian energy.** In addition to empirical observations, we further explore the influence of edge degree on the graph stability. Toward this, in the context of graph theory, we consider the graph Laplacian energy $E(\mathcal{G})$ mathematically defined as: $\sum_{n=1}^{N} |\lambda_n - 1|$ where $\{\lambda_n\}_{n=1}^{N}$ represents the spectrum of normalized graph Laplacian (Allem et al., 2016). Since all eigenvalues of the normalized graph Laplacian are bounded in $[0, 2]$, $E(\mathcal{G})$ measures how much the spectrum deviates from the median value 1. In our analysis, we investigate the quantity $\Delta_{ij} \coloneqq E(\mathcal{G}_{-ij}) - E(\mathcal{G})$ where $\mathcal{G}_{-ij} = (\mathcal{V}, \mathcal{E} - \{(i, j)\})$, that is, how the energy changes if we remove each edge $(i, j)$ for all $(i, j) \in \mathcal{E}$. If $\Delta_{ij} > 0$, it means that the subgraph $\mathcal{G}_{-ij}$ has higher energy than that of the original graph, thus it can be understood that removing edge $(i, j)$ makes the graph spectrally unstable. As depicted in Figure 2 for popular graph datasets, it is observed that the higher energy predominantly occurs when low-degree edges are eliminated. This observation substantiates the importance of preserving low-degree edges in terms of spectral stability.

**Theoretical evidence.** The very recent study (Tang & Liu, 2023) provides the theoretical understanding of model generalization for popular GNN families. More precisely, the upper bound for generalization error of each GNN family $\mathcal{F}$ relies on its Lipschitz constant, denoted by $L_{\mathcal{F}}$. In the same study, in case of GCN (the following argument is similar for other GNN families), $L_{\text{GCN}}$ is known to be dominantly governed by the norm $\|\boldsymbol{A}_{\text{sym}}\|_{\infty}$ rather than the node feature $\boldsymbol{X}$ or model parameter $\boldsymbol{\Theta}$ where $\boldsymbol{A}_{\text{sym}}$ is a symmetrically normalized adjacency matrix. Further, the norm $\|\boldsymbol{A}_{\text{sym}}\|_{\infty}$ can be upper-bounded by $\sqrt{\frac{\deg_{\max} + 1}{\deg_{\min} + 1}}$ where $\deg_{\max}$ and $\deg_{\min}$ represent the maximum and minimum node degree respectively. From this bound, removing low-degree edges would result in a larger generalization gap, which corroborates the importance of low-degree edges in theory.

## 5 TEDDY: TRIMMING EDGES WITH DEGREE-BASED DISCRIMINATION STRATEGY

Given the empirical and theoretical evidence on the importance of graph structures in Section 4, we introduce TEDDY, a novel framework for one-shot edge sparsification. Our method selectively prunes edges leveraging scores assigned to individual edges derived from degree information (Section 5.1), followed by parameter sparsification (Section 5.3) based on $\ell_0$ projection. The whole process is accomplished within a *single* training phase, eliminating the need for conventional iterative training to attain GLT.

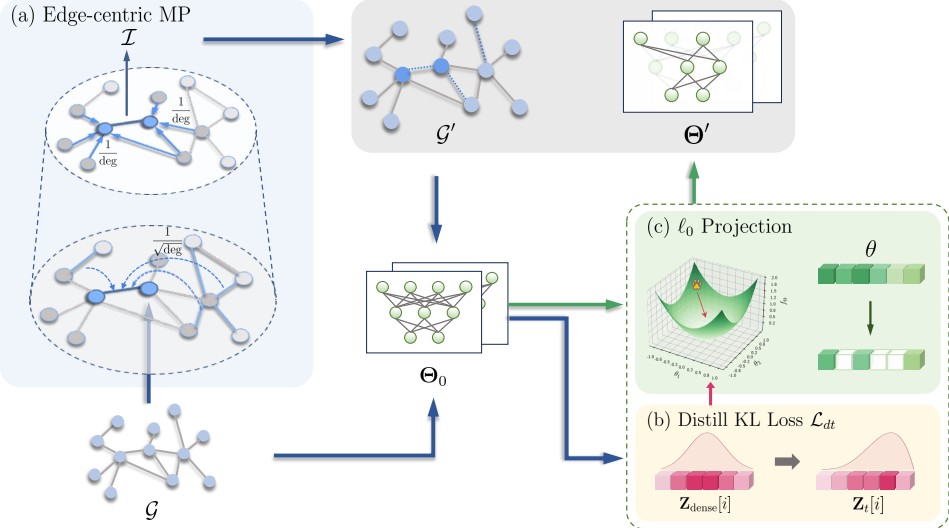

Figure 3: Overall framework for our graph sparsification TEDDY and parameter sparsification with projected gradient descent on $\ell_0$ ball.

## 5.1 GRAPH SPARSIFICATION

To integrate the significance of degree information discussed in Section 4, we design an edge-wise score in proportional to the inverse degree. The primary observation here is that relying solely on the degree of the corresponding node pair to prune high-degree edges does not sufficiently uncover the structural information pathways across multi-hop neighbors.

To see this more clearly, we consider a toy example in Figure 4. In this example, the edge degree of directly attributed nodes $i$ and $j$ is defined as $\deg_{1\text{-hop}}(e_{ij}) = (|\mathcal{N}_i| + |\mathcal{N}_j|)/2$, and the respective 2-hop edge degree is represented as $\deg_{2\text{-hop}}(e_{ij}) = (\mathbb{E}_{a\in\mathcal{N}_i}|\mathcal{N}_a| + \mathbb{E}_{b\in\mathcal{N}_j}|\mathcal{N}_b|)/2$. The most natural consideration of degrees is $\deg_{1\text{-hop}}$ to average the degrees

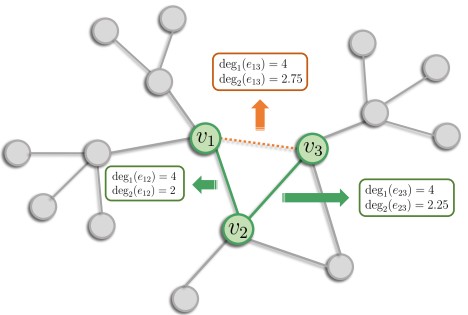

Figure 4: A toy example highlighting the importance of multi-level degree incorporation.

of the node pairs, but it fails to distinguish the relative importance of $e_{12}, e_{13}$, and $e_{23}$ in this example since they are all identical, *i.e.*, $\deg_{1\text{-hop}}(e_{12}) = \deg_{1\text{-hop}}(e_{13}) = \deg_{1\text{-hop}}(e_{23})$. As a result, $\deg_{1\text{-hop}}$-based score overlooks diverse message pathways through the common neighbor $v_2$ (colored in green) and 2-hop adjacent nodes (colored in gray) of $v_1$ and $v_3$. In contrast, the 2-hop edge degree allows for giving the priorities to edges $e_{12}$ and $e_{23}$ over $e_{13}$, *i.e.*, $\deg_{2\text{-hop}}(e_{12}) < \deg_{2\text{-hop}}(e_{23}) < \deg_{2\text{-hop}}(e_{13})$, facilitating a more refined edge score. This emphasizes the necessity to recognize degree information from a hierarchical perspective, culminating in a more nuanced score.

Inspired by this insight, we facilitate multi-level consideration of degree information within the message-passing (MP) algorithm, the core nature of contemporary GNNs. Our TEDDY adapts the original MP by injecting an edge-wise score within the MP. Toward this edge-wise score construction, we first define several quantities related to a node-wise score connected to individual edges. As an initial step, we adopt a monotonically decreasing function with respect to the node degree $\deg(v)$, defined by $g : \mathcal{V} \to \mathbb{R}$ that can be regarded as a type of score function for each node $v \in \mathcal{V}$. The monotonically decreasing characteristic of the function $g$ would reflect the importance of low-degree edges to some extent. As a design choice of $g$, there can be several candidates and we simply adopt $g(v) = 1/\sqrt{\deg(v)}$ in this paper. Equipped with this function $g$, TEDDY computes the importance of each node $v$ considering the average degree information of neighbors as below:

$$\bar{g}(v) = \frac{1}{|\mathcal{N}_v|} \sum_{u\in\mathcal{N}_v} g(u) \tag{3}$$

Following this, TEDDY divides $\overline{g}(v)$ by the degree of the node itself, represented as $\widetilde{g}(v)$, reflecting the significance of preserving edges with lower degrees:

$$\widetilde{g}(v) = \frac{\overline{g}(v)}{\deg(v)} \tag{4}$$

Let $\boldsymbol{g} \in \mathbb{R}^N$ denote the aggregated values of the function $g$ computed at all nodes $v \in \mathcal{V}$. Finally, our method attains edge-wise scores $\boldsymbol{T}_{edge}$ through the outer-product of $\widetilde{\boldsymbol{g}}$:

$$\boldsymbol{T}_{edge} = \widetilde{\boldsymbol{g}}\widetilde{\boldsymbol{g}}^\top \tag{5}$$

In practice, Eq. (3) is computed as $\widehat{\boldsymbol{A}}\boldsymbol{g} \in \mathbb{R}^N$, where $\widehat{\boldsymbol{A}} = \boldsymbol{D}^{-1}\boldsymbol{A}$ with $\boldsymbol{D}^{-1}$ being utilized to average the degree information of neighbors. Hence, Eq. (4) can be equivalently written as $\widetilde{\boldsymbol{g}} = \boldsymbol{D}^{-1}\widehat{\boldsymbol{A}}\boldsymbol{g} \in \mathbb{R}^N$. In this perspective, the computation of $\widetilde{\boldsymbol{g}}$ exhibits a form of edge-centric MP, which is depicted as well in Figure 5(b). As illustrated, the degree information of neighboring edges, $e_{ik}$, of the given edge $e_{ij}$ is propagated via proposed edge-wise score matrix $\boldsymbol{T}_{\text{edge}}$. At the same time, the self-information $1/\deg(i)$ is integrated during our edge-centric MP, analogous to standard MP across all nodes illustrated in (a).

Propagating the degree information, TEDDY effectively assigns lower scores to non-critical edges - those with a high priority for removal. These edges either possess high degree or are part of alternative multi-hop pathways for message propagation. Our method also demonstrates linear complexity in the number of nodes and edges, which will be detailed in Appendix A. Concurrently, our TEDDY aims to selectively preserve pivotal edges by considering low-degrees in a broader view, thereby maintaining essential information flow across the influential reception regions of individual nodes.

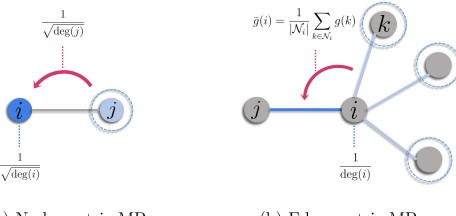

(a) Node-centric MP  (b) Edge-centric MP

Figure 5: Illustration of Edge-centric MP.

## 5.2 DISTILLATION FROM DENSE GNNS

To effectively maintain information in the original graph, we employ the distillation phase by matching the logits of the representation learned on the entire graph (Hinton et al., 2015). In our experience, we observe that the representation obtained from the entire graph plays an important role in model generalization. Let $\boldsymbol{Z}_{\text{dense}}$ and $\boldsymbol{Z}$ be the representation trained on the whole graph and the sparse subgraphs, respectively. Then, our final objective function would be mathematically

$$\mathcal{L}_{dt}(\mathcal{G}, \boldsymbol{\Theta}) := \mathcal{L}(\mathcal{G}, \boldsymbol{\Theta}) + \lambda_{dt}\mathbb{KL}\big(\text{softmax}(\boldsymbol{Z}/\tau), \text{softmax}(\boldsymbol{Z}_{\text{dense}}/\tau)\big), \tag{6}$$

where $\mathbb{KL}(p, q)$ represents the KL-divergence between two (possibly empirical) distributions $p(\cdot)$ and $q(\cdot)$. Further, $\lambda_{dt}$ controls the strength of the distillation and $\tau$ is a temperature parameter. For marginal hyperparameter tuning, we use a unit temperature $\tau = 1$ in practice.

## 5.3 PARAMETER SPARSIFICATION

The iterative nature of previous studies (Chen et al., 2021; You et al., 2022; Hui et al., 2023) on identifying sparse subnetworks necessitates an infeasible search space. The aforementioned studies require determining the number of pruning rounds and the per-round pruning ratio in advance to achieve target sparsity level. To alleviate such computational inefficiency, we prune the parameter connection within just single training with a sparsity-inducing optimization algorithm; projected gradient descent (PGD) on $\ell_0$ ball. In optimization literature, the projected gradient descent can be regarded as a special case of the proximal-type algorithm, which is known to enjoy rich properties both in theory and practice in the perspective of deep learning (Oymak, 2018; Yun et al., 2021a). Given target sparsity ratio $p_\theta \in [0, 1]$ for the model parameter, the expected number of non-zero entries would be $h := \lceil(1 - p_\theta)d\rceil$. Let $\mathcal{C}_h := \{\boldsymbol{\Theta} \in \mathbb{R}^d : \|\boldsymbol{\Theta}\|_0 = h\}$ be the $\ell_0$ ball for $h$-sparse parameter, then our parameter sparsification can be described only in a single line as

$$\boldsymbol{\Theta}_{t+1} = \text{proj}_{\mathcal{C}_h}\big(\boldsymbol{\Theta}_t - \eta\nabla_{\boldsymbol{\Theta}}\mathcal{L}_{dt}(\mathcal{G}, \boldsymbol{\Theta}_t)\big) \tag{7}$$

---

**Algorithm 1** TEDDY: **T**rimming **E**dges with **D**egree-based **D**iscrimination strateg**Y**

---

1: **Input:** Graph $\mathcal{G} = \{\boldsymbol{A}, \boldsymbol{X}\}$, GNN $f(\mathcal{G}, \boldsymbol{\Theta})$, target sparsity ratio $p_g, p_\theta \in [0, 1]$
2: **Initialize:** Initial model parameter $\boldsymbol{\Phi}_0, \boldsymbol{\Theta}_0 \in \mathbb{R}^d$ for pretraining and sparse training respectively
3: **procedure** PRETRAINING ON ENTIRE GRAPH
4:      **for** $t = 0, 1, \ldots, T - 1$ **do**
5:          $\boldsymbol{\Phi}_{t+1} \leftarrow \boldsymbol{\Phi}_t - \eta \nabla_{\boldsymbol{\Phi}} \mathcal{L}(\mathcal{G}, \boldsymbol{\Phi}_t)$
6:      **end for**
7:      Choose $\boldsymbol{\Phi}_* \in \{\boldsymbol{\Phi}_1, \cdots, \boldsymbol{\Phi}_t\}$ for best validation and get representation $\boldsymbol{Z}_{\text{dense}} \leftarrow f(\mathcal{G}, \boldsymbol{\Phi}_*)$
8: **end procedure**

9: **procedure** SEARCHING SUBGRAPH
10:      $\boldsymbol{A}' \leftarrow \boldsymbol{A}$ and $\mathcal{I} \leftarrow$ Smallest $\lceil p_g |\mathcal{E}| \rceil$ entries among edge index in $\boldsymbol{T}_{\text{edge}}$          ▷ Eq. (5)
11:      $\boldsymbol{A}'[\mathcal{I}] \leftarrow \boldsymbol{0}$ and obtain sparse subgraph $\mathcal{G}' \leftarrow \{\boldsymbol{A}', \boldsymbol{X}\}$
12: **end procedure**

13: **procedure** SEARCHING SUBNETWORK
14:      $\mathcal{C} \leftarrow \{\Theta \in \mathbb{R}^d : \|\Theta\|_0 = \lceil (1 - p_\theta)d \rceil\}$          ▷ $\ell_0$ ball for $\lceil (1 - p_\theta)d \rceil$-sparse parameter
15:      **for** $t = 0, 1, \ldots, T - 1$ **do**
16:          $\boldsymbol{Z}_t \leftarrow f(\mathcal{G}', \boldsymbol{\Theta}_t)$
17:          $\mathcal{L}_{dt}(\mathcal{G}', \boldsymbol{\Theta}_t) \leftarrow \mathcal{L}(\mathcal{G}', \boldsymbol{\Theta}_t) + \lambda_{dt} \mathbb{KL}\big(\text{softmax}(\boldsymbol{Z}_t), \text{softmax}(\boldsymbol{Z}_{\text{dense}})\big)$      ▷ Eq. (6)
18:          $\boldsymbol{\Theta}_{t+1} \leftarrow \text{proj}_{\mathcal{C}}\big(\boldsymbol{\Theta}_t - \eta \nabla_{\boldsymbol{\Theta}} \mathcal{L}_{dt}(\mathcal{G}', \boldsymbol{\Theta}_t)\big)$          ▷ Eq. (7)
19:      **end for**
20:      Choose $\boldsymbol{\Theta}' \in \{\boldsymbol{\Theta}_1, \cdots, \boldsymbol{\Theta}_T\}$ for best validation and obtain sparse subnetwork $f_{\boldsymbol{\Theta}'}$
21: **end procedure**

22: **Output:** Graph lottery tickets $\{\mathcal{G}', f_{\boldsymbol{\Theta}'}\}$.          ▷ Definition 1

---

where $\mathcal{L}_{dt}$ is defined in Eq. (6). The main advantage of PGD-based sparsification lies in its ability to encourage the desired level of sparsity without iterative processes, which enables us to rapidly identify graph lottery tickets. We summarize the detailed overall procedures in Algorithm 1 and Figure 3 illustrates our framework.

## 6 EXPERIMENTS

In our empirical studies, we primarily focus on semi-supervised node classification tasks, most regularly considered in graph representation learning. We consider two sets of experiments: (i) small-and medium-size problems (Section 6.1) and (ii) large-scale experiments (Section 6.2). We compare our TEDDY with original GNN performance (Vanilla), UGS (Chen et al., 2021) and WD-GLT (Hui et al., 2023) on regular-scale experiments (Section 6.1). Due to the computational time required for the Sinkhorn iterations in WD-GLT, the baseline WD-GLT is inevitably excluded for our large-scale experiments (Section 6.2). The GNN models, our proposed method, and the baselines are implemented upon PyTorch (Paszke et al., 2019), PyTorch Geometric (Fey & Lenssen, 2019), and OGB (Hu et al., 2020). The detailed experimental configurations are provided in Appendix B.

### 6.1 SMALL- AND MEDIUM-SCALE TASKS

In alignment with experiments in UGS, we evaluate the performance of our TEDDY on three benchmark datasets: Cora, Citeseer, and Pubmed (Sen et al., 2008) on three representative GNN architectures: GCN (Kipf & Welling, 2016), GIN (Xu et al., 2018a), and GAT (Veličković et al., 2017). Figure 6 illustrates the performance comparison with varying level of graph sparsity.

Although the main goal is to preserve the performance of vanilla dense GNN, our method improves the original accuracy in *more than half* of the considered settings across all sparsity levels. In particular, GIN on the Citeseer dataset shows the outstanding result (refer to the second row and third column in Figure 6). In this setting, the best accuracy is observed as 73.20%, at the graph sparsity $p_g = 43.12\%$ achieving a 5.4% improvement upon the vanilla performance. Moreover, our TEDDY consistently outperforms the baselines across all experimental settings with remarkable improvements in GAT. Note that, at the maximum graph sparsity, TEDDY enhances the performance in a range of $12.8\% \sim 20.4\%$ compared to the optimal performances of the baselines. In addition, we highlight MACs (Multiply-Accumulate operations) at inference for GIN as showcase examples. In Figure 7,

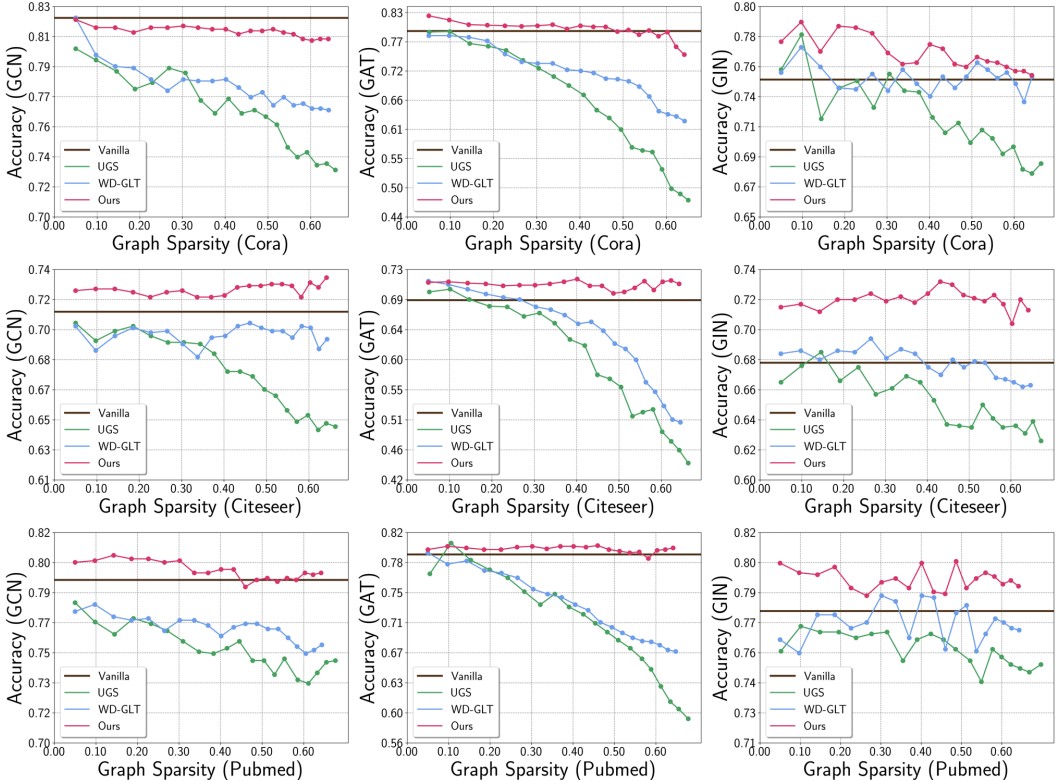

Figure 6: Experimental results on training GCN/GAT/GIN on Cora/Citeseer/Pubmed datasets.

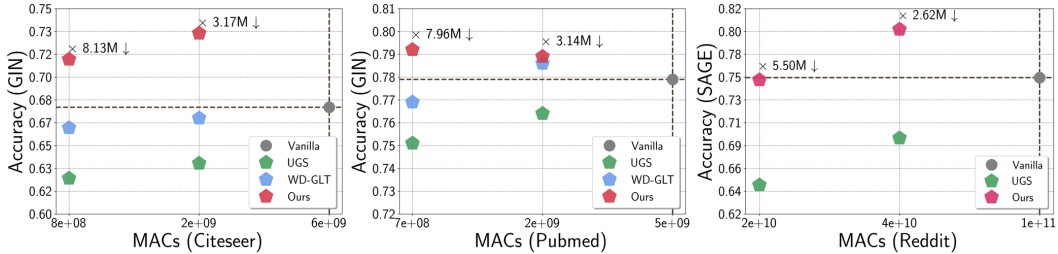

Figure 7: MAC comparisons for (**Left**) GIN on Citeseer (**Middle**) GIN on Pubmed (**Right**) SAGE on Reddit (large-scale experiment) as showcase examples.

GLTs found by TEDDY achieve dramatic savings of MACs (about 8 times smaller) on Citeseer and Pubmed, even with superior performance to the vanilla networks. Due to the space constraints, our state-of-the-art performance concerning the weight sparsity is deferred to Appendix A.

**Qualitative Analysis.** To demonstrate that our method proficiently incorporates the degree information inherent in the graph structure, we provide qualitative analysis concerning the average degrees of pruned edges for our TEDDY and baselines. As illustrated in Figure 8, our method effectively targets edges with high degrees across various graph sparsity levels. Moreover, the average degree of TEDDY continues to

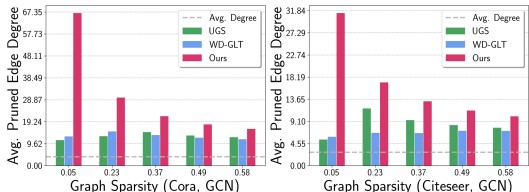

Figure 8: Qualitative analysis.

decrease as the sparsity ratio increases, which underscores that TEDDY prioritizes edges displaying low-degree information over multi-level adjacency considerations. On the contrary, the same phenomenon is not observed in UGS and WD-GLT, implying a lack of awareness of edge degrees.

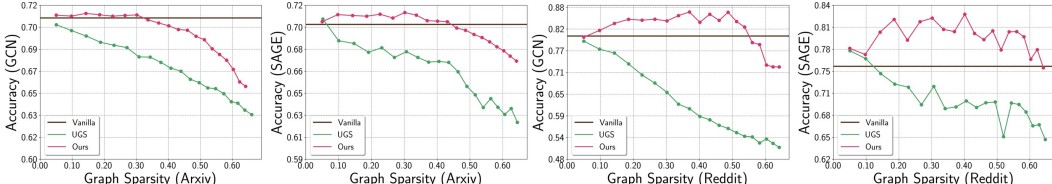

Figure 9: Experimental results on large-scale datasets, training GCN/SAGE on Arxiv/Reddit.

Table 1: Performance of proposed TEDDY (in percentage) on extremely sparse regimes, averaged over 5 runs.

| Simulations | Vanilla | 20-th | 25-th | 30-th | 35-th | 40-th |
|---|---|---|---|---|---|---|
| GS(%) | 0 | 64.14 | 72.25 | 78.53 | 83.38 | 87.14 |
| WS(%) | 0 | 64.15 | 72.26 | 78.54 | 83.39 | 87.15 |
| Cora | $76.34 \pm 0.79$ | $76.20 \pm 0.69$ | $\mathbf{76.64 \pm 0.76}$ | $\mathbf{77.38 \pm 0.97}$ | $\mathbf{77.20 \pm 1.12}$ | $\mathbf{76.82 \pm 1.00}$ |
| Citeseer | $68.10 \pm 0.77$ | $\mathbf{71.16 \pm 0.66}$ | $\mathbf{70.58 \pm 1.43}$ | $\mathbf{71.54 \pm 0.52}$ | $\mathbf{71.42 \pm 0.56}$ | $\mathbf{71.12 \pm 0.51}$ |
| Pubmed | $77.90 \pm 0.14$ | $\mathbf{79.70 \pm 0.26}$ | $\mathbf{79.36 \pm 0.71}$ | $\mathbf{79.68 \pm 0.37}$ | $\mathbf{80.48 \pm 0.50}$ | $\mathbf{80.98 \pm 0.42}$ |

**Quantitative Analysis.** We expand our investigation to assess the efficacy of our proposed method under extreme sparsity conditions. Employing GIN as a base architecture, we increase both the graph sparsity and the weight sparsity to a maximum of $87\%$ and conduct the same pruning procedure across 5 runs. According to Table 1, our TEDDY persists in enhancing classification accuracy even under harsh scenarios. Our method surpasses the original performance in all settings across all datasets except at graph sparsity and weight sparsity $p_g \approx p_\theta = 64.14\%$ (refer to 20-th simulations in Table 1). Notably, the performance of TEDDY exhibits a progressive improvement with the increment in sparsity ratio, marking $3.08\%$ advancement in comparison to the vanilla GIN on Pubmed.

## 6.2 LARGE-SCALE PROBLEMS

To further substantiate our analysis, we extend our experiments for two large-scale datasets: ArXiv (Hu et al., 2020) and Reddit (Zeng et al., 2019). Unlike in Section 6.1, we choose GCN and GraphSAGE (SAGE, Hamilton et al. (2017)) as backbone architectures since they are representative GNNs for these datasets. Displayed in Figure 9, TEDDY can relatively preserve the test accuracy compared to UGS. Specifically, in training SAGE on the Reddit, it is apparent that our TEDDY even surpasses the original performance of vanilla SAGE up to graph sparsity $p_g = 62.26\%$, which indicates the significant elevation (refer to the second last sparsity). The best performance in this setting is observed as $82.72\%$ at the graph sparsity $p_g = 40\%$, which shows $7.4\%$ and $12.4\%$ improvements upon vanilla SAGE and UGS respectively. Concurrently, TEDDY exhibits more advanced pruning performance upon UGS when it comes to the Arxiv dataset as well. Our method succeeds in attaining the GLT up to the graph sparsity $p_g = 33.66\%$ and $p_g = 45.96\%$ for GCN and SAGE, whereas UGS fails to discover GLT at this regime. Hence, these results imply that TEDDY can identify GLT with advanced performance regardless of graph size, which underscores its high versatility. Moreover, we accentuate MACs savings in large-scale experiments with SAGE on Reddit as our spotlight example. As can be seen in Figure 7, our TEDDY achieves 5.5 times smaller models maintaining the original performance while UGS shows significant degradation in this regime.

## 7 CONCLUSION

In this paper, we introduced TEDDY, the novel edge sparsification method that leverages structural information to preserve the integrity of principal information flow. Based on our observations on low-degree edges, we carefully designed an edge sparsification that effectively incorporates the graph structural information. After sparse subgraphs in hand, we efficiently induced the parameter sparsity via projected gradient descent on the $\ell_0$ ball with distillation. We also validated the superiority of TEDDY via extensive empirical studies and successfully identified sparser graph lottery tickets with significantly better generalization. As future work, we plan to investigate how to incorporate the node feature information into our TEDDY framework and explore the theoretical properties of TEDDY.

ACKNOWLEDGEMENT

This work was supported by Institute of Information & Communications Technology Planning & Evaluation (IITP) grant (No.2019-0-00075, Artificial Intelligence Graduate School Program (KAIST), No.2019-0-01371, Development of brain-inspired AI with human-like intelligence, No.2021-0-02068, Artificial Intelligence Innovation Hub) and the National Research Foundation of Korea (NRF) grants (No.2018R1A5A1059921) funded by the Korea government (MSIT). This work was also supported by Samsung Electronics Co., Ltd (No.IO201214-08133-01).

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

# SUPPLEMENTARY MATERIALS

# A FURTHER ANALYSES AND EXPERIMENTS

This section provides supplemental analyses and experiments to further validate the efficacy of TEDDY. We begin by examining the complexity of TEDDY in Section A.1, followed by an investigation of low-degree edges in other GNN architectures in Section A.2. Subsequently, we discuss the pruning performance relative to weight sparsity in Section A.3. Sections A.4 and A.5 extend our evaluation of TEDDY to include pruning on graph transformers and inductive node classification scenarios. A step-wise evaluation of our approach is detailed in Section A.6. Section A.7 focuses on experiments investigating sole graph sparsification under dense weight conditions, specifically assessing TEDDY's edge pruning capabilities against baselines, including the random pruning method in DropEdge (Rong et al., 2019). Finally, Sections A.8 and A.9 present a pairwise sparsity analysis and pruning time consumption metrics, respectively, to demonstrate the efficiency and effectiveness of TEDDY against comparative baselines.

## A.1 COMPLEXITY ANALYSIS

In practice, the computation of edge-wise scores in TEDDY requires $O(N + M)$ space complexity, since $\boldsymbol{T}_{edge} \in \mathbb{R}^M$ is only computed for the existing edges. More precisely, $g(v)$ is first computed by the row-wise summation of the sparse adjacency matrix. Then we compute $\overline{g}(v)$ via *sparse* matrix multiplication between degree vector and the adjacency matrix. Following this, $\widetilde{g}\widetilde{g}^\top$ is efficiently obtained by *element-wise* multiplication between $\widetilde{g}(v)$ of nodes $v$ in rows and $\widetilde{g}(u)$ of nodes $u$ in columns of the edge index. Hence, $\boldsymbol{T}_{edge}$ can be efficiently obtained with a complexity linear in the number of nodes and edges.

## A.2 LOW-DEGREE OBSERVATION ON OTHER GNN ARCHITECTURES

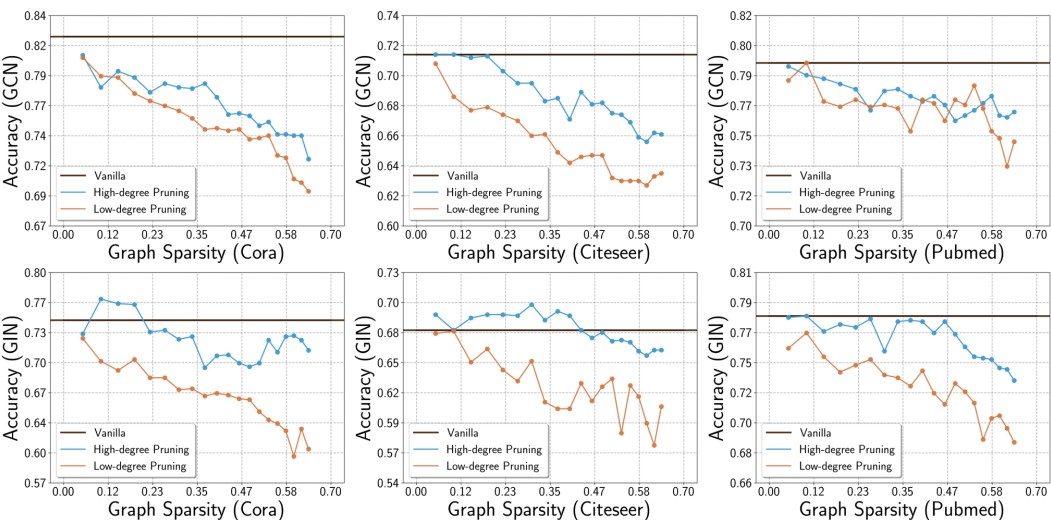

Figure 10: Low degree observations for GCN/GIN architectures on Cora/Citeseer/Pubmed datasets.

As an extension to Section 4, we compare the performance between the highest and the lowest degree edge elimination, leveraging GCN (Kipf & Welling, 2016) and GIN (Xu et al., 2018a). As depicted in Figure 10, removing low-degree edges consistently deteriorates the performance across all scenarios, especially for GIN trained on Cora dataset. The same trend is observed in the average performance of all configurations across multiple runs, as shown in Table 2.

## A.3 SPARSIFICATION PERFORMANCE RELATIVE TO WEIGHT SPARSITY

Here, we present the pruning performance of our TEDDY and baselines according to the weight sparsity, displayed in Figure 11 and 12. Consistent with the results related to graph sparsity, our method shows predominant performance compared to baselines across all evaluated settings, independent of the graph sizes.

Table 2: Performance on different edge degree pruning criteria (in percentage), averaged over 5 runs. $\text{deg}_{\text{low}}$ denotes low edge degree-based pruning, whereas $\text{deg}_{\text{high}}$ refers to high edge degree-based pruning.

| Simulations | | 1-st | | 5-th | | 10-th | | 15-th | | 20-th | |
|---|---|---|---|---|---|---|---|---|---|---|---|
| GNNs | Dataset | $\text{deg}_{\text{low}}$ | $\text{deg}_{\text{high}}$ | $\text{deg}_{\text{low}}$ | $\text{deg}_{\text{high}}$ | $\text{deg}_{\text{low}}$ | $\text{deg}_{\text{high}}$ | $\text{deg}_{\text{low}}$ | $\text{deg}_{\text{high}}$ | $\text{deg}_{\text{low}}$ | $\text{deg}_{\text{high}}$ |
| GCN | Cora | 80.38 ± 0.19 | **81.28 ± 0.64** | 77.66 ± 0.41 | **77.92 ± 0.34** | 75.36 ± 0.42 | **77.68 ± 0.32** | 74.24 ± 0.16 | **76.02 ± 0.43** | 69.54 ± 0.32 | **72.62 ± 0.32** |
| | Citeseer | 70.44 ± 0.21 | **70.88 ± 0.28** | 67.0 ± 0.27 | **70.60 ± 0.18** | 64.78 ± 0.31 | **67.58 ± 0.72** | 63.46 ± 0.42 | **67.72 ± 0.37** | 63.22 ± 0.51 | **65.70 ± 0.56** |
| | Pubmed | 78.32 ± 0.19 | **79.18 ± 0.12** | 76.66 ± 0.79 | **77.66 ± 0.27** | **77.52 ± 0.28** | 77.32 ± 0.20 | **77.72 ± 0.17** | 76.74 ± 0.10 | 75.20 ± 0.20 | **76.54 ± 0.08** |
| GAT | Cora | 77.64 ± 1.33 | **79.72 ± 1.06** | 68.34 ± 1.26 | **78.50 ± 1.16** | 56.72 ± 0.48 | **72.44 ± 1.10** | 50.12 ± 0.93 | **67.74 ± 0.92** | 44.78 ± 0.80 | **51.64 ± 1.67** |
| | Citeseer | 69.48 ± 1.36 | **70.96 ± 0.75** | 57.02 ± 0.64 | **66.86 ± 0.63** | 42.68 ± 2.14 | **59.74 ± 1.64** | 35.40 ± 1.50 | **49.86 ± 1.82** | 28.40 ± 0.97 | **43.24 ± 1.43** |
| | Pubmed | **78.40 ± 0.61** | 78.14 ± 1.01 | 71.66 ± 0.59 | **76.32 ± 0.55** | 58.48 ± 0.85 | **74.52 ± 0.56** | 49.52 ± 0.64 | **72.12 ± 0.64** | 44.10 ± 0.57 | **68.20 ± 0.61** |
| GIN | Cora | 74.86 ± 1.14 | **76.48 ± 1.66** | 70.70 ± 1.13 | **76.84 ± 1.81** | 66.70 ± 0.33 | **74.70 ± 2.32** | 62.72 ± 1.38 | **72.96 ± 0.55** | 58.82 ± 1.99 | **70.28 ± 0.99** |
| | Citeseer | 67.52 ± 0.99 | **68.78 ± 0.98** | 63.80 ± 1.48 | **69.16 ± 0.96** | 62.32 ± 1.14 | **68.46 ± 0.48** | 60.60 ± 1.27 | **66.92 ± 0.27** | 59.48 ± 2.25 | **65.64 ± 1.09** |
| | Pubmed | 75.64 ± 0.72 | **77.96 ± 0.36** | 75.12 ± 0.66 | **77.38 ± 0.27** | 73.68 ± 1.26 | **77.44 ± 0.10** | 70.74 ± 1.39 | **74.88 ± 0.62** | 70.88 ± 2.72 | **73.30 ± 0.11** |

## A.4 TEDDY WITH GRAPH TRANSFORMERS

To demonstrate the versatility of our method, we expand our experiments to encompass recent Transformer-based GNN architectures, UniMP (Shi et al., 2020), NAGphormer (Chen et al., 2022), and Specformer (Bo et al., 2023). Note that differentiable mask approach from baselines may prune edges more than pre-defined ratio, due to the possibility of multiple mask elements having the same value. Moreover, UGS (Chen et al., 2021) and WD-GLT (Hui et al., 2023) are infeasible to implement on Specformer and NAGphormer, since the differentiable mask for the adjacency matrix requires backpropagation through eigenvectors and eigenvalues utilized in these architectures, introducing a practical challenge with high complexity of $O(N^3)$ per iteration.

As depicted in Figure 13 and 14, TEDDY accomplishes stable pruning performance, surpassing all baselines when equipped with UniMP as a backbone. In particular, UGS and WD-GLT show significant degradation with severe unstability as the sparsity increases, whereas the performance of TEDDY is stable across all benchmark datasets. Our method also yields decent performance in NAGphormer and Specformer, notably in NAGphormer on the Cora dataset, with considerable performance enhancement over the original result. Overall, these results demonstrate TEDDY's versatility across diverse foundational architectures.

## A.5 TEDDY ON INDUCTIVE SEMI-SUPERVISED SETTINGS

We present additional experiments to further evaluate TEDDY's adaptability on *inductive* semi-supervised node classification task, where the fraction of the validation and test set is unobserved and larger than that of the training set. We modified the Cora, Citeseer, and Pubmed datasets (Sen et al., 2008) to a 20/40/40% split for training, validation, and testing phases, ensuring that the model had no prior exposure to the validation and test nodes during training.

The results, illustrated in Figure 15 and 16, demonstrate that TEDDY achieves prominent pruning performance on inductive setting as well. This is especially pronounced on the Cora and Pumbed datasets with GIN and, surpassing the accuracy of the vanilla GIN. Moreover, our method consistently achieves stable performance throughout all simulations on the Pubmed dataset, regardless of the base architectures. These findings strongly affirm TEDDY's ability to effectively generalize from smaller to larger graphs.

## A.6 ABLATION STUDY

To confirm that each component in our framework contributes individually to the edge/parameter sparsification, we conduct comprehensive ablation studies. Towards this, we provide a step-wise assessment result on Cora dataset, equipped with GAT (Veličković et al., 2017) as a base architecture. As illustrated in Figure 17, while integrating the distillation loss $\mathcal{L}_{dt}$ does improve the performance of the baselines, they still fail to match the performance of our method, even ours without $\mathcal{L}_{dt}$. This highlights a significant impact of proposed edge-centric message passing with degree statistics. Furthermore, our approach, which considers multiple levels of degree information (specified as Ours), consistently outperforms methods that rely solely on degree information from direct node pairs (specified as 1-hop Degree).

Further ablation studies were conducted to examine the impact of different components, as shown in Figure 18. These studies included: (1) TEDDY with all components, (2) TEDDY without distillation loss, (3) TEDDY with magnitude-based weight mask pruning, (4) UGS with all components, (5) UGS with distillation loss, (6) UGS with $\ell_0$-based projected gradient descent (PGD). Analogous to the previous findings, UGS enhanced with $\mathcal{L}_{dt}$, remains to be suboptimal compared to TEDDY without $\mathcal{L}_{dt}$ across all benchmark datasets. Meanwhile, the performance of UGS degrades dramatically when incorporating $\ell_0$ PGD. Intriguingly, TEDDY maintains robust performance with magnitude-based weight pruning, highlighting its flexibility across varying weight

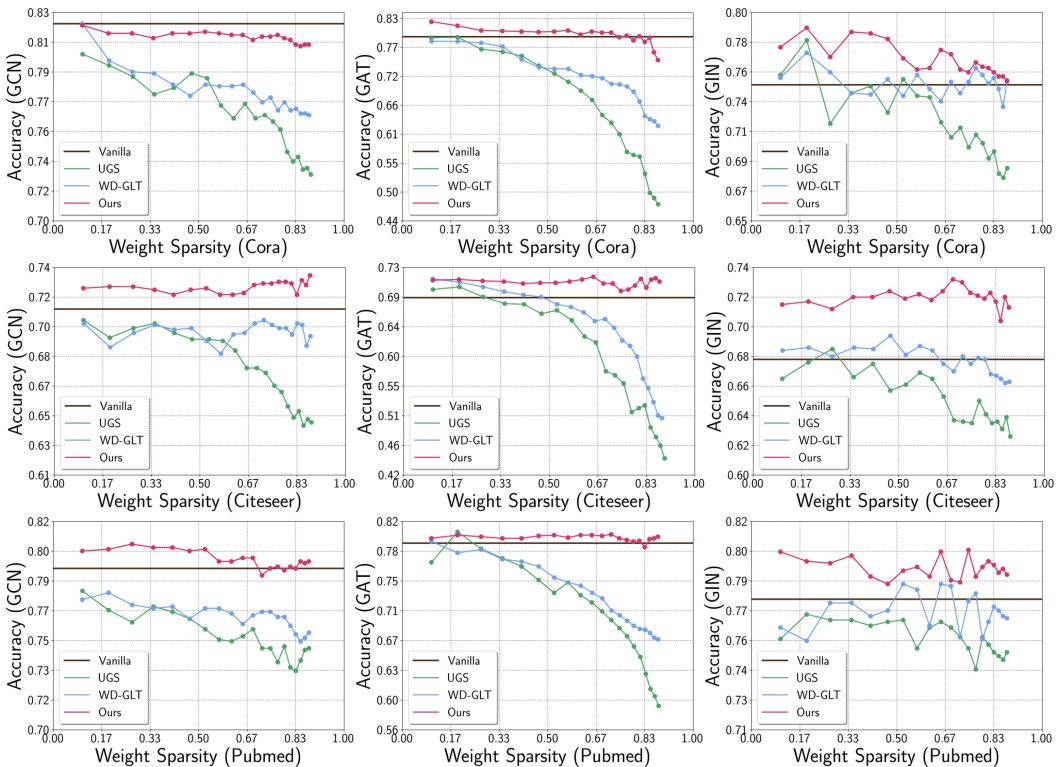

Figure 11: Additional experimental results in Section 6.1 with respect to the weight sparsity.

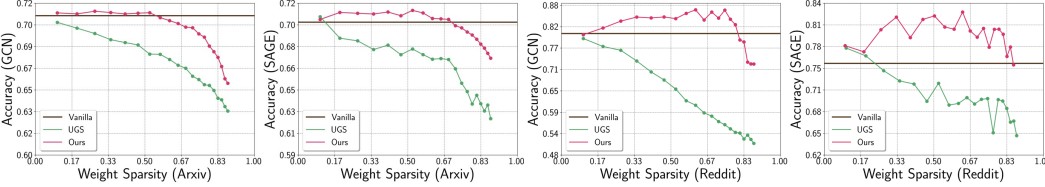

Figure 12: Additional experimental results in Section 6.2 with respect to the weight sparsity.

sparsification strategies. Nevertheless, leveraging PGD offers an optimal balance of efficiency and effectiveness, obviating the need for additional iterative training to obtain lottery tickets.

### A.7 EXCLUSIVE EDGE SPARSIFICATION PERFORMANCE

In this experiment, we focused exclusively on graph sparsification to directly compare the effects of various edge pruning techniques, including the random dropping strategy employed in DropEdge (Rong et al., 2019), with our TEDDY. The results, as shown in Figure 19, indicate that TEDDY consistently outperforms existing baseline methods, including DropEdge. Notably, our approach not only maintains but also enhances the accuracy of vanilla GNNs on the Citeseer and Pubmed datasets across all benchmarked GNNs. Furthermore, the figure highlights that the use of the DropEdge technique leads to suboptimal performance, emphasizing the importance of a more sophisticated pruning methodology.

### A.8 PAIRWISE SPARSITY ANALYSIS

To demonstrate the stability of TEDDY, we present the results in Table 1 as a heatmap in Figure 20. The performance in each heatmap element is an average performance over 5 random seeds. Overall, the performance of TEDDY is not uniformly affected by increased sparsity. In the Cora dataset, our method demonstrates resilience to higher graph sparsity levels, surpassing the vanilla GIN's accuracy of 76.34%. This suggests that a sparser graph generated from our method has a potential to mitigate overfitting achieve generalization. For the Citeseer dataset, the performance remains relatively stable across a range of weight sparsity levels, indicating a degree of robustness to parameter reduction. Furthermore, all configurations exceed the vanilla GIN's accuracy of 68.1%,

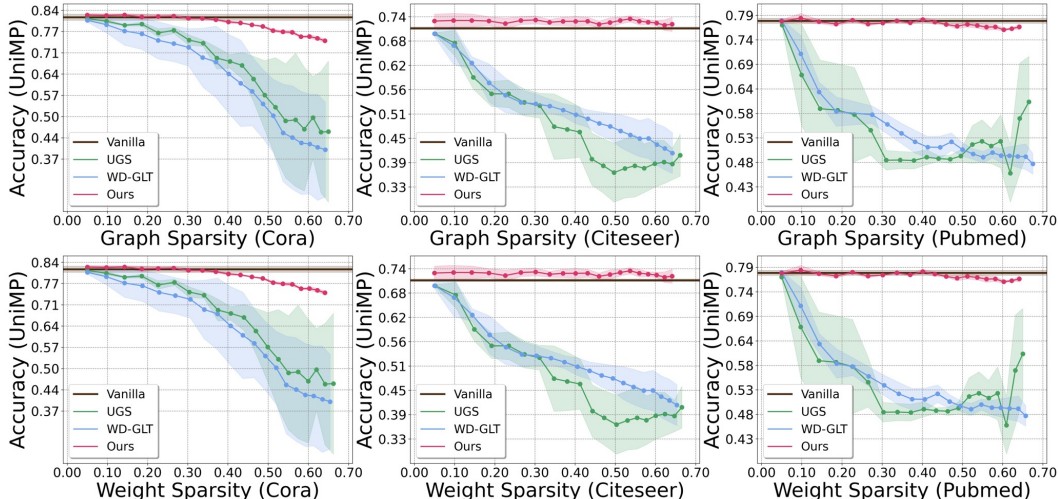

Figure 13: Average classification performance of training UniMP on Cora/Citeseer/Pubmed datasets.

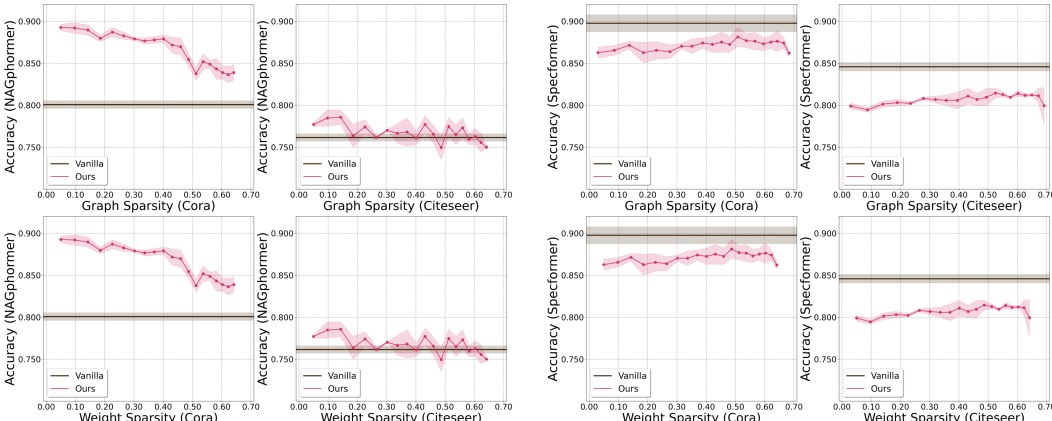

Figure 14: Average classification performance of training NAGphormer (left) /Specformer (right) on Cora/Citeseer datasets.

strongly indicating the effectiveness of TEDDY across diverse sparsities. The similar efficacy is observed in the Pubmed dataset, where all elements surpasses the vanilla GIN's performance of 77.9%. Interestingly, we observe a trend in the Pubmed where the performance generally improves with increased sparsity, with the highest accuracy achieved under the most extreme sparsity ($p_g = p_\theta = 85\%$).

## A.9 COMPARISON OF PRUNING PROCESS DURATION

This subsection details an wall-clock time analysis of (1) TEDDY with all components and (2) Projected Gradient Descent (PGD), the weight sparsification method in TEDDY, to verify its efficiency in comparison to the baselines. We reported the time consumption of our method and baselines for each distinct simulation, average over five runs. As demonstrated in Table 3, the results clearly substantiates that TEDDY consistently achieves a significant reduction in pruning time relative to the baselines. This is especially pronounced in GIN on the Pubmed dataset, where TEDDY outperforms WD-GLT by a remarkable margin, displaying a maximum time saving of 220.72 seconds during the 15-th simulation. Similar trend is observed in the results in large-scale datasets, depicted in Table 4 and 5, where TEDDY's time consumption is nearly half that of UGS. The maximum duration gap is revealed in the Reddit dataset, with TEDDY concluding the last simulation 131.84 seconds quicker than UGS in GCN. Note that the duration of WD-GLT is reported as N/A due to prohibitable computational time for the Sinkhorn iteration.

Following the comparison of wall-clock times for the overall process, we further investigate how efficient our projected gradient descent on the $\ell_0$-ball is in terms of actual runtime compared to iterative approaches. For fair comparison, we use the original dense adjacency matrix without graph sparsification while pruning only model parameters. Table 6 ∼ 8 illustrate our wall-clock time comparison for citation networks, Arxiv, and Reddit

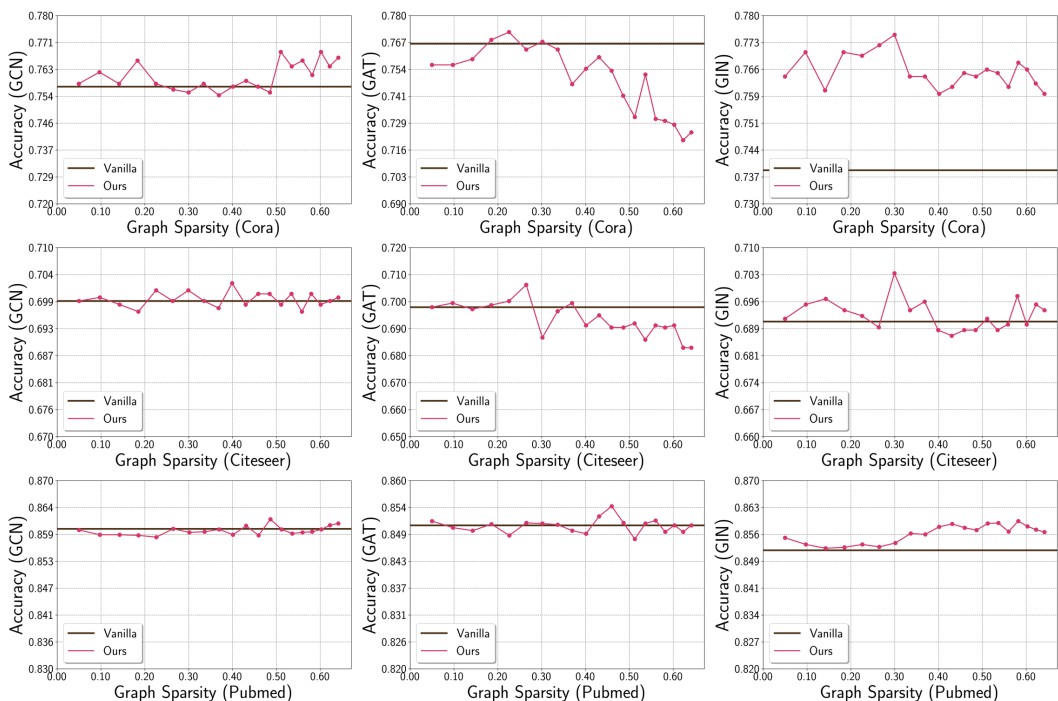

Figure 15: Inductive experimental results on training GCN/GAT/GIN on Cora/Citeseer/Pubmed datasets, with respect to the graph sparsity.

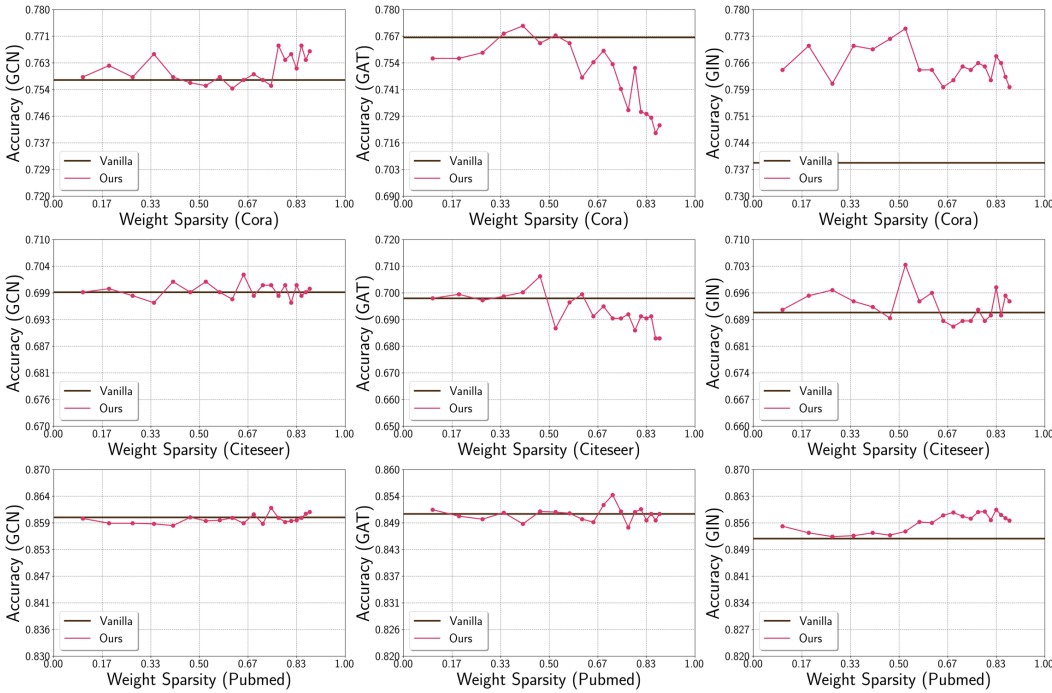

Figure 16: Inductive experimental results on training GCN/GAT/GIN on Cora/Citeseer/Pubmed datasets, with respect to the weight sparsity.

datasets. As similar to the overall process depicted in Table 3 ∼ 5, TEDDY consistently saves the actual time about more than 2 times upon baselines across most of the considered settings. More notably, for large-scale dataset Arxiv and Reddit, the wall-clock time gap become markedly pronounced. This is might be due to the fact that the iterative approaches additionally introduce a larger number of learning parameters (for parameter

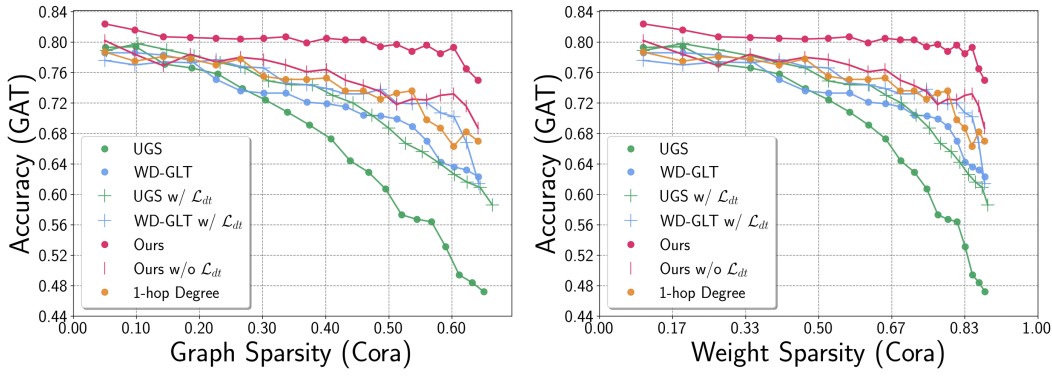

Figure 17: Ablation study on GAT architectures trained on Cora dataset.

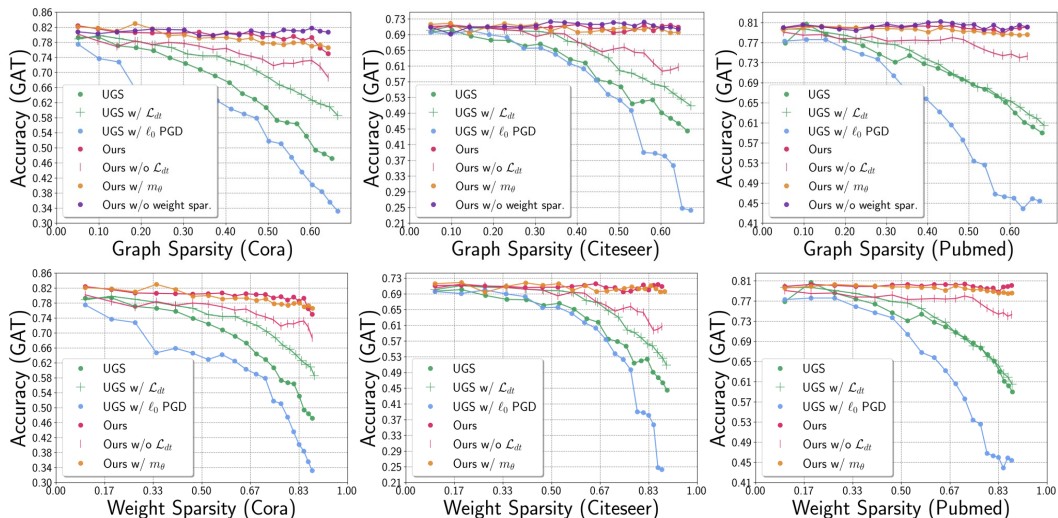

Figure 18: Ablation study on GAT trained on Cora/Citeseer/Pubmed dataset, across diverse configurations.

masks) particularly when dealing with large-scale datasets. As in overall pruning process, duration of WD-GLT is reported as N/A or OOM due to prohibitive computation incurred by Sinkhorn iterations.

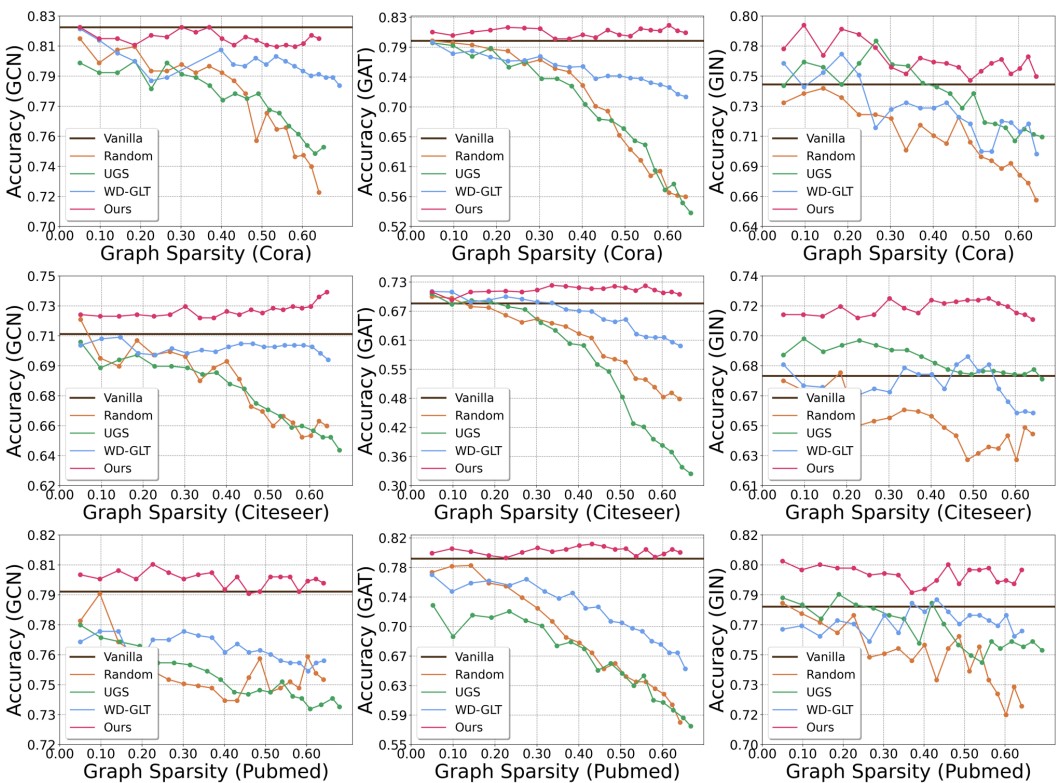

Figure 19: Experimental results for a sole graph sparsification including DropEdge (represented as Random) on Cora/Citeseer/Pubmed datasets, equipped with GCN/GAT/GIN.

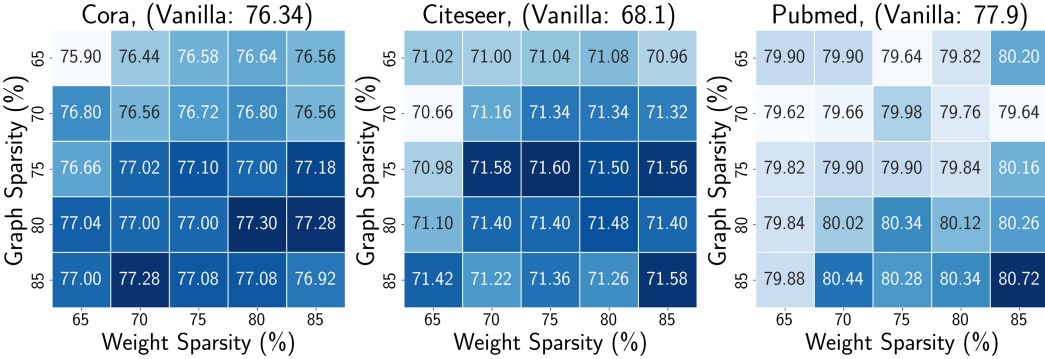

Figure 20: Performance on diverse extreme sparsity combinations (in percentage) of TEDDY with GIN, averaged over 5 runs.

Table 3: Duration (sec) of the **overall** pruning process for each simulation on Cora/Citeseer/Pubmed datasets, averaged over 5 runs. The comparison is conducted on the machine with NVIDIA Titan Xp and Intel(R) Xeon(R) CPU E5-2630 v4 @ 2.20GHz.

| Cora | | 1-st | 5-th | 10-th | 15-th | 20-th |
|---|---|---|---|---|---|---|
| GCN | UGS | $6.30_{\pm 0.55}$ | $5.84_{\pm 0.39}$ | $5.57_{\pm 0.80}$ | $5.65_{\pm 0.59}$ | $5.56_{\pm 0.73}$ |
| | WD-GLT | $54.11_{\pm 1.28}$ | $53.61_{\pm 0.39}$ | $53.59_{\pm 0.17}$ | $53.52_{\pm 0.23}$ | $53.77_{\pm 0.37}$ |
| | TEDDY | $\mathbf{2.12}_{\pm 0.08}$ | $\mathbf{2.15}_{\pm 0.16}$ | $\mathbf{2.09}_{\pm 0.12}$ | $\mathbf{2.11}_{\pm 0.13}$ | $\mathbf{2.18}_{\pm 0.05}$ |
| GAT | UGS | $7.58_{\pm 0.44}$ | $7.11_{\pm 0.51}$ | $6.97_{\pm 0.59}$ | $7.03_{\pm 0.60}$ | $7.26_{\pm 0.24}$ |
| | WD-GLT | $83.81_{\pm 10.02}$ | $79.98_{\pm 8.09}$ | $80.76_{\pm 6.30}$ | $80.84_{\pm 6.29}$ | $79.40_{\pm 4.81}$ |
| | TEDDY | $\mathbf{3.00}_{\pm 0.07}$ | $\mathbf{2.96}_{\pm 0.14}$ | $\mathbf{3.04}_{\pm 0.08}$ | $\mathbf{2.93}_{\pm 0.40}$ | $\mathbf{3.01}_{\pm 0.05}$ |
| GIN | UGS | $4.38_{\pm 0.81}$ | $4.11_{\pm 0.29}$ | $4.11_{\pm 0.28}$ | $4.06_{\pm 0.30}$ | $4.01_{\pm 0.19}$ |
| | WD-GLT | $68.55_{\pm 10.13}$ | $72.43_{\pm 10.91}$ | $70.76_{\pm 10.45}$ | $72.04_{\pm 10.23}$ | $72.67_{\pm 11.89}$ |
| | TEDDY | $\mathbf{1.86}_{\pm 0.25}$ | $\mathbf{1.91}_{\pm 0.31}$ | $\mathbf{1.85}_{\pm 0.28}$ | $\mathbf{1.86}_{\pm 0.32}$ | $\mathbf{1.86}_{\pm 0.27}$ |
| Citeseer | | 1-st | 5-th | 10-th | 15-th | 20-th |
| GCN | UGS | $8.21_{\pm 1.72}$ | $7.44_{\pm 0.52}$ | $7.55_{\pm 0.53}$ | $7.36_{\pm 0.55}$ | $7.33_{\pm 0.42}$ |
| | WD-GLT | $83.73_{\pm 16.75}$ | $78.24_{\pm 14.09}$ | $77.67_{\pm 19.85}$ | $70.75_{\pm 22.62}$ | $69.51_{\pm 17.12}$ |
| | TEDDY | $\mathbf{2.65}_{\pm 0.33}$ | $\mathbf{2.72}_{\pm 0.29}$ | $\mathbf{2.70}_{\pm 0.34}$ | $\mathbf{2.71}_{\pm 0.29}$ | $\mathbf{2.80}_{\pm 0.34}$ |
| GAT | UGS | $8.78_{\pm 0.84}$ | $8.09_{\pm 0.52}$ | $8.47_{\pm 0.36}$ | $8.64_{\pm 0.77}$ | $8.41_{\pm 0.64}$ |
| | WD-GLT | $93.96_{\pm 20.92}$ | $86.23_{\pm 19.97}$ | $89.55_{\pm 22.03}$ | $89.03_{\pm 22.84}$ | $86.74_{\pm 21.02}$ |
| | TEDDY | $\mathbf{3.21}_{\pm 0.57}$ | $\mathbf{3.35}_{\pm 0.46}$ | $\mathbf{3.51}_{\pm 0.50}$ | $\mathbf{3.55}_{\pm 0.49}$ | $\mathbf{3.63}_{\pm 0.51}$ |
| GIN | UGS | $6.61_{\pm 1.11}$ | $6.44_{\pm 0.77}$ | $6.05_{\pm 0.38}$ | $5.98_{\pm 0.38}$ | $6.16_{\pm 0.42}$ |
| | WD-GLT | $108.74_{\pm 3.78}$ | $109.59_{\pm 4.25}$ | $106.69_{\pm 3.24}$ | $109.91_{\pm 4.16}$ | $107.93_{\pm 3.35}$ |
| | TEDDY | $\mathbf{2.51}_{\pm 0.10}$ | $\mathbf{2.53}_{\pm 0.11}$ | $\mathbf{2.46}_{\pm 0.09}$ | $\mathbf{2.49}_{\pm 0.05}$ | $\mathbf{2.52}_{\pm 0.16}$ |
| Pubmed | | 1-st | 5-th | 10-th | 15-th | 20-th |
| GCN | UGS | $8.40_{\pm 0.54}$ | $7.80_{\pm 0.31}$ | $7.61_{\pm 0.55}$ | $7.95_{\pm 0.27}$ | $8.12_{\pm 0.30}$ |
| | WD-GLT | $176.48_{\pm 26.20}$ | $161.88_{\pm 31.51}$ | $160.96_{\pm 34.56}$ | $141.58_{\pm 9.83}$ | $142.68_{\pm 12.39}$ |
| | TEDDY | $\mathbf{3.23}_{\pm 0.20}$ | $\mathbf{3.27}_{\pm 0.13}$ | $\mathbf{3.16}_{\pm 0.14}$ | $\mathbf{3.14}_{\pm 0.21}$ | $\mathbf{3.08}_{\pm 0.12}$ |
| GAT | UGS | $9.85_{\pm 0.91}$ | $9.35_{\pm 0.59}$ | $9.16_{\pm 0.72}$ | $9.61_{\pm 0.67}$ | $9.41_{\pm 0.75}$ |
| | WD-GLT | $69.64_{\pm 8.76}$ | $68.69_{\pm 8.31}$ | $66.07_{\pm 5.49}$ | $64.38_{\pm 2.74}$ | $63.66_{\pm 1.11}$ |
| | TEDDY | $\mathbf{3.90}_{\pm 0.26}$ | $\mathbf{3.82}_{\pm 0.18}$ | $\mathbf{3.58}_{\pm 0.20}$ | $\mathbf{3.63}_{\pm 0.21}$ | $\mathbf{3.50}_{\pm 0.30}$ |
| GIN | UGS | $7.42_{\pm 1.01}$ | $7.17_{\pm 0.41}$ | $7.11_{\pm 0.51}$ | $7.03_{\pm 0.49}$ | $6.95_{\pm 0.41}$ |
| | WD-GLT | $218.87_{\pm 6.40}$ | $214.61_{\pm 7.44}$ | $213.63_{\pm 5.83}$ | $223.74_{\pm 4.99}$ | $216.23_{\pm 6.28}$ |
| | TEDDY | $\mathbf{3.20}_{\pm 0.18}$ | $\mathbf{3.12}_{\pm 0.13}$ | $\mathbf{3.06}_{\pm 0.10}$ | $\mathbf{3.02}_{\pm 0.07}$ | $\mathbf{2.97}_{\pm 0.12}$ |

Table 4: Duration of the **overall** pruning process for each simulation on Arxiv dataset, averaged over 5 runs. The comparison is conducted on the machine with NVIDIA GeForce RTX 3090 and Intel(R) Xeon(R) Gold 5215 CPU @ 2.50GHz.

| Arxiv | | 1-st | 5-th | 10-th | 15-th | 20-th |
|---|---|---|---|---|---|---|
| GCN | UGS | $82.28_{\pm 19.32}$ | $77.93_{\pm 13.14}$ | $75.26_{\pm 12.89}$ | $73.18_{\pm 11.94}$ | $71.50_{\pm 10.81}$ |
| | WD-GLT | N/A | N/A | N/A | N/A | N/A |
| | TEDDY | $\mathbf{43.45}_{\pm 0.02}$ | $\mathbf{39.70}_{\pm 0.04}$ | $\mathbf{36.10}_{\pm 0.05}$ | $\mathbf{33.35}_{\pm 0.03}$ | $\mathbf{31.16}_{\pm 0.02}$ |
| SAGE | UGS | $74.22_{\pm 3.98}$ | $73.37_{\pm 2.61}$ | $76.18_{\pm 4.86}$ | $75.18_{\pm 5.07}$ | $70.94_{\pm 1.00}$ |
| | WD-GLT | N/A | N/A | N/A | N/A | N/A |
| | TEDDY | $\mathbf{45.10}_{\pm 0.02}$ | $\mathbf{41.96}_{\pm 0.02}$ | $\mathbf{38.85}_{\pm 0.02}$ | $\mathbf{36.43}_{\pm 0.01}$ | $\mathbf{34.58}_{\pm 0.02}$ |

Table 5: Duration of the **overall** pruning process for each simulation on Reddit dataset, averaged over 5 runs. The comparison is conducted on the machine with NVIDIA GeForce RTX 3090 and Intel(R) Xeon(R) Gold 5215 CPU @ 2.50GHz.

| Reddit | | 1-st | 5-th | 10-th | 15-th | 20-th |
|---|---|---|---|---|---|---|
| GCN | UGS | $180.80_{\pm 69.56}$ | $176.25_{\pm 54.61}$ | $158.97_{\pm 52.54}$ | $154.12_{\pm 42.40}$ | $155.35_{\pm 49.46}$ |
| | WD-GLT | OOM | OOM | OOM | OOM | OOM |
| | TEDDY | $\mathbf{47.68}_{\pm 0.02}$ | $\mathbf{40.29}_{\pm 0.03}$ | $\mathbf{32.92}_{\pm 0.02}$ | $\mathbf{27.56}_{\pm 0.01}$ | $\mathbf{23.51}_{\pm 0.10}$ |
| SAGE | UGS | $174.47_{\pm 34.80}$ | $173.08_{\pm 31.65}$ | $167.41_{\pm 32.09}$ | $166.55_{\pm 30.85}$ | $161.88_{\pm 30.77}$ |
| | WD-GLT | OOM | OOM | OOM | OOM | OOM |
| | TEDDY | $\mathbf{65.53}_{\pm 0.03}$ | $\mathbf{55.77}_{\pm 0.03}$ | $\mathbf{46.25}_{\pm 0.03}$ | $\mathbf{39.12}_{\pm 0.05}$ | $\mathbf{33.45}_{\pm 0.04}$ |

Table 6: Comparisons of duration (sec) of the **parameter** sparsification process for each method for each simulation on Cora/Citeseer/Pubmed datasets, averaged over 5 runs. For fair comparison, we consider only the parameter sparsification with original dense graph (i.e. without graph sparsification). The comparison is conducted on the machine with NVIDIA Titan Xp and Intel(R) Xeon(R) CPU E5-2630 v4 @ 2.20GHz.

| **Cora** | | **1-st** | **5-th** | **10-th** | **15-th** | **20-th** |
|---|---|---|---|---|---|---|
| | UGS | $6.09 \pm 0.81$ | $5.42 \pm 0.22$ | $5.71 \pm 0.47$ | $5.47 \pm 0.29$ | $5.51 \pm 0.22$ |
| GCN | WD-GLT | $62.78 \pm 3.06$ | $60.89 \pm 0.96$ | $61.38 \pm 0.93$ | $61.82 \pm 1.15$ | $61.92 \pm 0.87$ |
| | TEDDY | $\mathbf{2.39} \pm \mathbf{0.06}$ | $\mathbf{2.43} \pm \mathbf{0.04}$ | $\mathbf{2.44} \pm \mathbf{0.10}$ | $\mathbf{2.41} \pm \mathbf{0.04}$ | $\mathbf{2.41} \pm \mathbf{0.04}$ |
| | UGS | $6.00 \pm 1.04$ | $5.58 \pm 0.34$ | $5.60 \pm 0.29$ | $5.50 \pm 0.18$ | $5.48 \pm 0.19$ |
| GAT | WD-GLT | $61.95 \pm 3.52$ | $61.45 \pm 2.13$ | $61.40 \pm 1.90$ | $59.99 \pm 1.48$ | $60.27 \pm 2.29$ |
| | TEDDY | $\mathbf{2.43} \pm \mathbf{0.09}$ | $\mathbf{2.47} \pm \mathbf{0.09}$ | $\mathbf{2.50} \pm \mathbf{0.15}$ | $\mathbf{2.45} \pm \mathbf{0.08}$ | $\mathbf{2.44} \pm \mathbf{0.08}$ |
| | UGS | $4.33 \pm 0.74$ | $4.15 \pm 0.46$ | $4.05 \pm 0.18$ | $3.99 \pm 0.29$ | $3.87 \pm 0.20$ |
| GIN | WD-GLT | $61.98 \pm 3.15$ | $60.63 \pm 0.39$ | $60.38 \pm 0.46$ | $60.46 \pm 0.30$ | $60.42 \pm 0.45$ |
| | TEDDY | $\mathbf{1.76} \pm \mathbf{0.03}$ | $\mathbf{1.80} \pm \mathbf{0.05}$ | $\mathbf{1.82} \pm \mathbf{0.04}$ | $\mathbf{1.81} \pm \mathbf{0.02}$ | $\mathbf{1.81} \pm \mathbf{0.01}$ |
| **Citeseer** | | **1-st** | **5-th** | **10-th** | **15-th** | **20-th** |
| | UGS | $8.84 \pm 2.27$ | $7.87 \pm 1.16$ | $7.56 \pm 0.81$ | $7.56 \pm 0.62$ | $7.43 \pm 0.65$ |
| GCN | WD-GLT | $56.97 \pm 2.45$ | $56.05 \pm 0.86$ | $55.96 \pm 1.33$ | $55.38 \pm 0.60$ | $55.29 \pm 0.53$ |
| | TEDDY | $\mathbf{3.27} \pm \mathbf{0.04}$ | $\mathbf{3.37} \pm \mathbf{0.06}$ | $\mathbf{3.46} \pm \mathbf{0.03}$ | $\mathbf{3.50} \pm \mathbf{0.01}$ | $\mathbf{3.60} \pm \mathbf{0.05}$ |
| | UGS | $8.42 \pm 1.17$ | $7.95 \pm 0.58$ | $7.91 \pm 0.70$ | $7.89 \pm 0.49$ | $7.77 \pm 0.58$ |
| GAT | WD-GLT | $57.67 \pm 2.29$ | $56.90 \pm 0.57$ | $56.47 \pm 0.80$ | $57.30 \pm 1.17$ | $56.51 \pm 0.57$ |
| | TEDDY | $\mathbf{3.35} \pm \mathbf{0.05}$ | $\mathbf{3.39} \pm \mathbf{0.04}$ | $\mathbf{3.50} \pm \mathbf{0.03}$ | $\mathbf{3.55} \pm \mathbf{0.01}$ | $\mathbf{3.66} \pm \mathbf{0.05}$ |
| | UGS | $8.28 \pm 1.00$ | $8.37 \pm 1.01$ | $8.18 \pm 0.84$ | $8.16 \pm 0.95$ | $7.81 \pm 0.64$ |
| GIN | WD-GLT | $57.50 \pm 2.02$ | $56.11 \pm 0.26$ | $56.19 \pm 0.54$ | $56.88 \pm 1.40$ | $56.38 \pm 0.71$ |
| | TEDDY | $\mathbf{3.42} \pm \mathbf{0.05}$ | $\mathbf{3.46} \pm \mathbf{0.04}$ | $\mathbf{3.57} \pm \mathbf{0.04}$ | $\mathbf{3.62} \pm \mathbf{0.02}$ | $\mathbf{3.71} \pm \mathbf{0.05}$ |
| **Pubmed** | | **1-st** | **5-th** | **10-th** | **15-th** | **20-th** |
| | UGS | $9.21 \pm 0.96$ | $8.48 \pm 0.46$ | $8.66 \pm 0.62$ | $8.28 \pm 0.20$ | $8.38 \pm 0.42$ |
| GCN | WD-GLT | $202.28 \pm 1.98$ | $201.13 \pm 1.54$ | $200.82 \pm 2.08$ | $201.07 \pm 1.53$ | $201.16 \pm 1.22$ |
| | TEDDY | $\mathbf{3.91} \pm \mathbf{0.08}$ | $\mathbf{3.88} \pm \mathbf{0.07}$ | $\mathbf{3.90} \pm \mathbf{0.11}$ | $\mathbf{3.89} \pm \mathbf{0.10}$ | $\mathbf{3.91} \pm \mathbf{0.12}$ |
| | UGS | $13.69 \pm 0.97$ | $13.18 \pm 0.23$ | $13.05 \pm 0.24$ | $13.07 \pm 0.21$ | $13.12 \pm 0.14$ |
| GAT | WD-GLT | $66.03 \pm 0.75$ | $65.68 \pm 0.06$ | $65.65 \pm 0.12$ | $65.66 \pm 0.14$ | $65.62 \pm 0.09$ |
| | TEDDY | $\mathbf{6.33} \pm \mathbf{0.05}$ | $\mathbf{6.32} \pm \mathbf{0.03}$ | $\mathbf{6.33} \pm \mathbf{0.04}$ | $\mathbf{6.32} \pm \mathbf{0.04}$ | $\mathbf{6.33} \pm \mathbf{0.02}$ |
| | UGS | $8.68 \pm 0.78$ | $8.38 \pm 0.15$ | $8.37 \pm 0.19$ | $8.35 \pm 0.11$ | $8.26 \pm 0.07$ |
| GIN | WD-GLT | $157.18 \pm 0.48$ | $157.03 \pm 1.12$ | $156.43 \pm 0.93$ | $157.75 \pm 0.59$ | $157.88 \pm 0.51$ |
| | TEDDY | $\mathbf{4.07} \pm \mathbf{0.04}$ | $\mathbf{4.04} \pm \mathbf{0.03}$ | $\mathbf{4.06} \pm \mathbf{0.01}$ | $\mathbf{4.04} \pm \mathbf{0.04}$ | $\mathbf{4.04} \pm \mathbf{0.06}$ |

Table 7: Duration of the **parameter** sparsification process for each simulation on Arxiv dataset, averaged over 5 runs. The comparison is conducted on the machine with NVIDIA Titan Xp and Intel(R) Xeon(R) CPU E5-2630 v4 @ 2.20GHz.

Table 8: Duration of the **parameter** sparsification process for each simulation on Reddit dataset, averaged over 5 runs. The comparison is conducted on the machine with NVIDIA Titan Xp and Intel(R) Xeon(R) CPU E5-2630 v4 @ 2.20GHz.

| **Arxiv** | | **1-st** | **5-th** | **10-th** | **15-th** | **20-th** |
|---|---|---|---|---|---|---|
| | UGS | $130.66 \pm 15.32$ | $125.78 \pm 13.97$ | $128.90 \pm 14.12$ | $113.48 \pm 11.06$ | $119.41 \pm 11.84$ |
| GCN | WD-GLT | N/A | N/A | N/A | N/A | N/A |
| | TEDDY | $\mathbf{62.68} \pm \mathbf{1.29}$ | $\mathbf{57.10} \pm \mathbf{1.33}$ | $\mathbf{51.75} \pm \mathbf{0.97}$ | $\mathbf{47.55} \pm \mathbf{1.21}$ | $\mathbf{45.07} \pm \mathbf{1.20}$ |
| | UGS | $111.71 \pm 7.45$ | $104.29 \pm 9.21$ | $106.31 \pm 8.13$ | $94.02 \pm 7.78$ | $90.71 \pm 6.59$ |
| SAGE | WD-GLT | N/A | N/A | N/A | N/A | N/A |
| | TEDDY | $\mathbf{64.03} \pm \mathbf{0.16}$ | $\mathbf{64.03} \pm \mathbf{0.09}$ | $\mathbf{64.44} \pm \mathbf{0.17}$ | $\mathbf{64.13} \pm \mathbf{0.07}$ | $\mathbf{64.34} \pm \mathbf{0.13}$ |

| **Reddit** | | **1-st** | **5-th** | **10-th** | **15-th** | **20-th** |
|---|---|---|---|---|---|---|
| | UGS | $210.36 \pm 51.92$ | $200.75 \pm 43.13$ | $197.71 \pm 47.39$ | $173.81 \pm 39.13$ | $170.48 \pm 42.65$ |
| GCN | WD-GLT | OOM | OOM | OOM | OOM | OOM |
| | TEDDY | $\mathbf{70.34} \pm \mathbf{0.05}$ | $\mathbf{65.69} \pm \mathbf{0.11}$ | $\mathbf{58.43} \pm \mathbf{0.03}$ | $\mathbf{51.72} \pm \mathbf{0.15}$ | $\mathbf{47.03} \pm \mathbf{0.13}$ |
| | UGS | $203.75 \pm 29.72$ | $194.58 \pm 25.91$ | $188.02 \pm 31.74$ | $181.86 \pm 35.12$ | $182.97 \pm 26.94$ |
| SAGE | WD-GLT | OOM | OOM | OOM | OOM | OOM |
| | TEDDY | $\mathbf{80.63} \pm \mathbf{0.11}$ | $\mathbf{78.21} \pm \mathbf{0.06}$ | $\mathbf{67.93} \pm \mathbf{0.13}$ | $\mathbf{63.11} \pm \mathbf{0.04}$ | $\mathbf{51.03} \pm \mathbf{0.16}$ |

# B EXPERIMENTAL SETTINGS

## B.1 DATASET STATISTICS

Table 9: Statistics of benchmark datasets.

| Dataset | #Nodes | #Edges | #Classes | #Features | Split ratio |
|---------|--------|--------|----------|-----------|-------------|
| Cora | 2,708 | 5,429 | 7 | 1,433 | 120/500/1000 |
| Citeseer | 3,327 | 4,732 | 6 | 3,703 | 140/500/1000 |
| Pubmed | 19,717 | 44,338 | 3 | 500 | 60/500/1000 |
| Arxiv | 169,343 | 1,166,243 | 40 | 128 | 54%/18%/28% |
| Reddit | 232,965 | 23,213,838 | 41 | 602 | 66%/10%/24% |

Table 9 provides comprehensive statistics of the datasets used in our experiments, including the number of nodes, edges, classes, and features.

## B.2 IMPLEMENTATION DETAILS

We implement GNN models and our proposed TEDDY using PyTorch Paszke et al. (2019) and PyTorch Geometric Fey & Lenssen (2019). The experiments are conducted on an RTX 2080 Ti (11GB) and RTX 3090 (24GB) GPU machines. Following the experiments in UGS (Chen et al., 2021), we consider the same experiment settings on Cora, Citeseer, and Pubmed dataset across all GNN architectures, except for GAT whose hidden dimension is set to 64 owing to the suboptimal performance of vanilla GAT. Regarding the experiments on large-scale datasets, we employ three-layer and two-layer GNN on Arxiv and Reddit, respectivly, while fixing the hidden dimension as 256 across both GCN and SAGE. Analogous to the regular-scale experiment, we select the Adam optimizer with an initial learning rate of 0.01 and weight decay as 0 uniformly across all large-scale settings. We adopted per-simulation pruning ratio as $p_g = p_\theta = 0.05$ and hyperparameter search space for $\mathcal{L}_{dt}$ within the range of $[0.01, 200]$. The source code for our experiments is available at https://github.com/hyunjin72/TEDDY.

## B.3 IMPLEMENTATION DISCREPANCY OF BASELINES WITH GAT

During our experiments, we observe an implementation discrepancy in the baselines (Chen et al., 2021; Hui et al., 2023) when interfaced with GAT. Specifically, at the attention phase, the edge mask to eliminate edges in the adjacency matrix is applied preceding the softmax operation on attention coefficients, yielding non-zero outputs even for pruned edges. In effect, the edges are not expected to be actually pruned. To rectify this inconsistency, we slightly revise the source code to guarantee the actual removal of designated edges during the forward pass of GAT.

## B.4 REPRODUCING WD-GLT (HUI ET AL., 2023)

Since there is no official implementations in public, we reproduce the baseline results in Hui et al. (2023) by ourselves. In order for our paper to be self-contained, we introduce the technique in Hui et al. (2023) in this section. For better generalization of GLT, Hui et al. (2023) propose a novel regularization based on Wasserstein distance (WD) between different classes. Toward this, let $\boldsymbol{Z} := f(\mathcal{G}, \boldsymbol{\Theta})$ be the representation obtained from GNN. Further, we define $\boldsymbol{Z}^c$ and $\boldsymbol{Z}^{\bar{c}}$ as

$$\boldsymbol{Z}^c := \{\boldsymbol{z}_i \in \text{row}(\boldsymbol{Z}) : \text{argmax}(\boldsymbol{z}_i) = c\}, \quad \boldsymbol{Z}^{\bar{c}} := \{\boldsymbol{z}_i \in \text{row}(\boldsymbol{Z}) : \text{argmax}(\boldsymbol{z}_i) \neq c\}$$

The authors maximize the Wasserstein distance between $\boldsymbol{Z}^c$ and $\boldsymbol{Z}^{\bar{c}}$ which is defined by

$$\text{WD}(\boldsymbol{Z}^c, \boldsymbol{Z}^{\bar{c}}) := \inf_{\pi \sim \Pi(\boldsymbol{Z}^c, \boldsymbol{Z}^{\bar{c}})} \mathbb{E}_{(\boldsymbol{z}_i, \boldsymbol{z}_j) \sim \pi} \big[\|\boldsymbol{z}_i - \boldsymbol{z}_j\|_2\big] \tag{8}$$

where $\Pi(\boldsymbol{Z}^c, \boldsymbol{Z}^{\bar{c}}$ is the set of all joint distributions $\pi(\boldsymbol{z}_i, \boldsymbol{z}_j)$ whose marginals are $\boldsymbol{Z}^c$ and $\boldsymbol{Z}^{\bar{c}}$ respectively. The authors compute this WD for all classes $c \in \mathcal{C} = \{1, 2, \cdots, C\}$, thus the regularization term would be

$$\mathcal{R}(\boldsymbol{m}_g, \boldsymbol{m}_\theta, \boldsymbol{\Theta}) = -\sum_{c \in \mathcal{C}} \text{WD}(\boldsymbol{Z}^c, \boldsymbol{Z}^{\bar{c}}) \tag{9}$$

The Wasserstein distance between two empirical distributions can be estimated by solving the entropy-regularized Wasserstein distance with Sinkhorn iterations. Together with the loss function $\mathcal{L}$ in Eq. (2), WD-GLT consider the final objective $\widetilde{\mathcal{L}} = \mathcal{L} + \lambda \mathcal{R}$ with regularization coefficient $\lambda$.

Equipped with the objective in Eq. (9), Hui et al. (2023) solve the minmax optimization problem for robustness formulated as

$$\min_{\boldsymbol{m}_\theta, \boldsymbol{\Theta}} \max_{\boldsymbol{m}_g} \widetilde{\mathcal{L}}(\boldsymbol{m}_g, \boldsymbol{m}_\theta, \boldsymbol{\Theta}) \tag{10}$$

The optimization has two procedures: (i) adversarially perturbing the mask $\boldsymbol{m}_g$ via gradient ascent and (ii) minimizing the objective $\mathcal{L}$ with respect to $\boldsymbol{m}_\theta$ and $\boldsymbol{\Theta}$ via gradient descent. More precisely, the update rule would be

$$\boldsymbol{m}_g^{(t+1)} = \text{proj}_{[0,1]^{N \times N}} \left( \boldsymbol{m}_g^{(t)} + \eta_1 \nabla_{\boldsymbol{m}_g} \widetilde{\mathcal{L}}(\boldsymbol{m}_g^{(t)}, \boldsymbol{m}_\theta^{(t)}, \boldsymbol{\Theta}^{(t)}) \right) \tag{11}$$

$$\boldsymbol{m}_\theta^{(t+1)} = \boldsymbol{m}_\theta^{(t)} - \eta_2 \Big( \nabla_{\boldsymbol{m}_\theta} \widetilde{\mathcal{L}}(\boldsymbol{m}_g^{(t+1)}, \boldsymbol{m}_\theta^{(t)}, \boldsymbol{\Theta}^{(t)}) + \alpha \underbrace{\left( \frac{\partial \boldsymbol{m}_g^{(t+1)}}{\partial \boldsymbol{m}_\theta} \right)^{\mathsf{T}} \nabla_{\boldsymbol{m}_g} \widetilde{\mathcal{L}}(\boldsymbol{m}_g^{(t+1)}, \boldsymbol{m}_\theta^{(t)}, \boldsymbol{\Theta}^{(t)})}_{\text{Implicit gradient by Eq. (11)}} \Big) \tag{12}$$

$$\boldsymbol{\Theta}^{(t+1)} = \boldsymbol{\Theta}^{(t)} - \eta_2 \Big( \nabla_{\boldsymbol{\Theta}} \widetilde{\mathcal{L}}(\boldsymbol{m}_g^{(t+1)}, \boldsymbol{m}_\theta^{(t)}, \boldsymbol{\Theta}^{(t)}) + \alpha \underbrace{\left( \frac{\partial \boldsymbol{m}_g^{(t+1)}}{\partial \boldsymbol{\Theta}} \right)^{\mathsf{T}} \nabla_{\boldsymbol{m}_g} \widetilde{\mathcal{L}}(\boldsymbol{m}_g^{(t+1)}, \boldsymbol{m}_\theta^{(t)}, \boldsymbol{\Theta}^{(t)})}_{\text{Implicit gradient by Eq. (11)}} \Big) \tag{13}$$

where $\eta_1$ and $\eta_2$ are learning rates for $\boldsymbol{m}_g$ and $(\boldsymbol{m}_\theta, \boldsymbol{\Theta})$ respectively, and $\alpha$ controls the strength of the implicit gradient by chain rule. The projection operator onto $[0, 1]^{N \times N}$ ensures that each entry in the graph mask $\boldsymbol{m}_g$ has its value in the interval $[0, 1]$. Hui et al. (2023) report that $\eta_1 = \eta_2 = 0.01$ and $\alpha = 0.1$ are used for experiments. The update rules in can be also found in Equation (9) $\sim$ (11) in Hui et al. (2023).

We make several remarks on solving the minmax optimization in Eq. (10) from our experiences.

- In fact, the only case of $\alpha = 1$ corresponds to the correct update rule for the minmax optimization in Eq. (10).

- In terms of implementations, the cases for $\alpha = 0$ and $\alpha = 1$ are easy to implement. For example, in PyTorch library, when solving the inner maximization problem, we only turn on or off the `create_graph` option in `loss.backward()` where $\alpha = 0$ corresponds to `False` and $\alpha = 1$ is for `True`. The other case $\alpha \neq 0, 1$ might require non-trivial handling in backpropagation with Sinkhorn iterations multiple times, which could be computationally infeasible.

- Hence, we could not consider the case $\alpha = 0.1$ which is the reported hyperparameter in Hui et al. (2023). However, we conisder the cases of $\alpha = 0$ and $\alpha = 1$. For both cases of $\alpha = 0$ and $\alpha = 1$, we observe that the values of each entry of $\boldsymbol{m}_g$ always tends to increase, hence the projection operator $\text{proj}_{[0,1]^{N \times N}}$ makes all the entries of $\boldsymbol{m}_g$ by 1. Therefore, the magnitude-based graph edge sparsification could not select suitable coordinates to be pruned.

For such reasons, we revise the optimization problem in Eq. (11) as

$$\min_{\boldsymbol{m}_g, \boldsymbol{m}_\theta, \boldsymbol{\Theta}} \widetilde{\mathcal{L}}(\boldsymbol{m}_g, \boldsymbol{m}_\theta, \boldsymbol{\Theta}) \tag{14}$$

which just minimize the objective with respect to all trainable parameters (specifically, for updating $\boldsymbol{m}_g$, we keep the projection operator $\text{proj}_{[0,1]^{N \times N}}$). Under the revised optimization, we only require just single Sinkhorn iterations for approximating Wasserstein distance and successfully reproduce the similar results in Hui et al. (2023).

