# OpenReview forum: "TEDDY: Trimming Edges with Degree-based Discrimination Strategy"
_ICLR.cc/2024/Conference — ICLR 2024 poster_

### Official Review · Reviewer_QTmY · 2023-10-27

**Soundness:** 3 good
**Presentation:** 2 fair
**Contribution:** 2 fair
**Rating:** 5
**Confidence:** 5

**Summary:**

This paper introduces an intuitively derived graph sparsification technique based on edge degrees, and integrates network distillation techniques for weight sparsification. The pruning process is independent of IMP and operates in a one-shot manner.

**Strengths:**

S1. This work demonstrates that high degree edges are less important via empirical observations, which is  significant in LTH community.

S2. Although many prior efforts have explored one-shot GLT; nevertheless, this paper seems to well strike the balance between efficiency and performance.

**Weaknesses:**

W1. The authors have avoided discussions concerning complexity, yet it appears that $T_{edge}$ might necessitate an $O(N^2)$ space complexity. Any comments on this?

W2. This work appears to be a fusion of two research lines:
- From the perspective of graph sparsification, there have already been efforts to prune edges based on edge properties and graph connectivity. The assertions made in \cite{wang2022pruning} closely align with this study, suggesting that the removal of "non-bridge" edges (corresponding to the "low-degree edges") has minimal impact on graph information flow. Furthermore, it provides theoretical support and error bound analysis.
- From the perspective of weight sparsification, [1, 2] have extensively explored the feasibility of using PGD for parameter pruning. I cannot see the substantial differences or new contributions in this paper.

W3. While the innovations are intuitively appealing, they still lack theoretical substantiation. The assertion that high-degree edges should be pruned is not a trivial conclusion. I find the similar claim in [3], which examines graph connectivity, to be more appealing.

[1] Zhou X, Zhang W, Xu H, et al. Effective sparsification of neural networks with global sparsity constraint[C]//Proceedings of the IEEE/CVF Conference on Computer Vision and Pattern Recognition. 2021: 3599-3608.
[2] Cai X, Yi J, Zhang F, et al. Adversarial structured neural network pruning[C]//Proceedings of the 28th ACM International Conference on Information and Knowledge Management. 2019: 2433-2436.
[3] Wang L, Huang W, Zhang M, et al. Pruning graph neural networks by evaluating edge properties[J]. Knowledge-Based Systems, 2022, 256: 109847.
[4] Hui B, Yan D, Ma X, et al. Rethinking Graph Lottery Tickets: Graph Sparsity Matters[J]. arXiv preprint arXiv:2305.02190, 2023.
[5] Chen T, Sui Y, Chen X, et al. A unified lottery ticket hypothesis for graph neural networks[C]//International conference on machine learning. PMLR, 2021: 1695-1706.

**Questions:**

The baseline performance depicted in Fig 6 and 11 appears highly questionable, displaying a significant drop compared to the results reported in [4, 5], as well as my previous replications. Perhaps the authors should consider presenting a more rigorously established and fair baseline performance.

The references can be found in the weakness part.

**Details Of Ethics Concerns:**

N/A.

---

> ### Author Response · Authors · 2023-11-21
> **Response to Reviewer QTmY (1/4)**
>
> We sincerely appreciate the reviewer’s helpful and constructive feedbacks. We would like to address the reviewer’s concerns as below.
>
> **[On Complexity]**
>
> **W1.** In practice, $T_{edge} \in |\mathcal{E}|$ is only computed for the existing edges, hence it requires only $O(|\mathcal V|+|\mathcal{E}|)=O(N+M)$ space complexity. The computation of $T_{edge}$ involves (1) degree computation for individual nodes by row-wise sparse summation of the adjacency matrix, (2) Mean-aggregating the degree vector with (sparse) adjacency matrix to obtain $\widetilde g$ in a sparse matrix multiplication manner, and (3) Element-wise multiplication between $\widetilde g(v)$ of nodes $v$ in rows (i.e., source nodes) and $\widetilde g(u)$ of nodes $u$ (i.e., destination nodes) in columns of the edge index. For better understanding, we attached the code snippet for computing $T_{edge}$ in the supplementary material (`compute_edge_score` function in `pruning.py`).

---

> ### Author Response · Authors · 2023-11-21
> **Response to Reviewer QTmY (2/4)**
>
> **[On Graph Sparsification]**
>
> **W2-1.** We appreciate your observation in drawing parallels between our work and the approach presented in [1]. However, we respectfully disagree with the view that our work, TEDDY, is closely related to [1].
>
> There exist several **clear differences and superiority** of TEDDY from the methodology proposed in [1], which we believe are worthwhile to clarify:
>
> 1. **No Direct Conceptual Alignment**: Although [1] demonstrates the edge importance via spectral analysis, the concept of the bridge is **NOT strictly aligned** with our high-degree/low-degree edges. According to the Definition 2 in the paper [1], the bridge refers to an edge whose deletion increases the number of connected components in the graph, and the reverse corresponds to the non-bridge. However, the bridge in the paper does not correspond to neither low-degree edge nor high-degree edge, since there could be still a chance of the existence of common neighbors between node pairs of both high- and low-degree edges, or the between the receptive fields (multi-hop neighbors).
> 2. **Training Pipeline Distinction**: More importantly, the primary distinction between our method and [1] is in the training procedure. In practice, PGEP method proposed in [1] prunes two types of edges in a fully trained network: (i) edge $(i,j)$ with different **predicted labels** $\widehat{y}_i\neq \widehat{y}_j$ (specified as negative edge) and (ii) non-bridge edges. Hence, TEDDY is the first approach to achieve effective pruning by utilizing **solely the structural information** of graph data.
>
> Furthermore, the aforementioned approach in [1] faces several practical issues. Firstly, to identify all non-bridge edges in the entire graph, one needs to execute graph algorithms such as DFS for *each node*, which can be **highly time-consuming** when it comes to large-scale datasets. Secondly, even if all non-bridge edges are identified, if the total number of non-bridge edges and negative edges is less than the target sparsity ratio, PGEP fails to achieve the desired pruning level. In consequence, this method may **not** be **available** in **high sparsity regimes**. On the other hand, given sparsity ratio for both graph and parameter, TEDDY can exactly achieve the target sparsity level in a more efficient way. Unfortunately, due to this cumbersome training procedure, we were not able to conduct the additional evaluation on [1] during the rebuttal period. Nevertheless, we will include the experiment in the final version.
>
> Thus, while there are superficial similarities in the definition, conceptual alignment and underlying training procedure of TEDDY are markedly different from [1]. We believe these differences are significant and contribute to the distinctiveness and novelty of our work in the realm of GNN pruning.
>
> ---
>
> **[On Parameter Sparsification]**
>
> **W2-2.** The projected gradient descent (PGD) is a well-known optimization algorithm with rich properties that has been used in various machine learning literature for a long time. The characteristics of PGD depend on the appropriate setting of the constraint set. In this regard, the two mentioned works [2, 3] indeed utilize projected gradient descent, but each has a different constraint set from ours. In the case of [2], projection is performed onto the $\ell_1$-ball, while [3] utilizes a constraint set satisfying the RIP-like condition. Note that both [2, 3] introduce additional hyperparameters, and tuning these hyperparameters is inevitable to reach the target sparsity level. In contrast, our approach directly projects onto the $\ell_0$-ball ***without introducing additional hyperparameters***, making it a novel attempt in sparsification.

---

> ### Author Response · Authors · 2023-11-21
> **Response to Reviewer QTmY (3/4)**
>
> **[On Theoretical Substantiation]**
>
> **W3.** As we discuss in **[Graph Sparsification]**, [1] introduce the concepts of bridge/non-bridge edges using graph spectral analysis, upon which the authors propose graph sparsification methods, but it is important to note that the notion of bridge/non-bridge edges in [1] does not entirely align with the low-degree/high-degree edges in our paper. As we note in **Theoretical evidence** of remarks in Section 4, the generalization error bounds of several GNN families heavily depends on the adjacency matrix rather than model parameter or node features. For the representative example, the error bound for GCN relies on the $\Vert A_{\text{sym}}\Vert_{\infty}$ for the symmetrically normalized adjacency matrix $A_{\text{sym}}$ and this norm can be further upper-bounded by $\sqrt{\frac{deg_{max}+1}{deg_{min}+1}}$. Note also that PGEP method introduced in [1] would be **computationally intractable** in practice since it should execute multiple times of graph search algorithm (such as DFS) for each node to identify bridge/non-bridge edges through a whole graph, which would be more problematic in a large-scale scenario. On the other hand, our approach TEDDY can practically identify to-be-pruned edges in a more computationally efficient way (please refer to the performance evaluation and time consumption on large-scale datasets in Section 6.2 and Appendix A.8, respectively).

---

> ### Author Response · Authors · 2023-11-21
> **Response to Reviewer QTmY (4/4)**
>
> **[Baseline Performance]**
>
> **Q1.** We believe that our experiment is conducted upon the fair setting. To demonstrate this, we have attached the source code for both our TEDDY and the baselines in supplementary.
>
> It's important to note the issue of the official implementation of UGS [4] on GAT (https://github.com/VITA-Group/Unified-LTH-GNN/blob/main/NodeClassification/gnns/gat_layer.py). According to the code, edge masking is applied *prior to* the softmax normalization of attention coefficients. However, this implementation leads to a notable issue where pruned edges inevitably receive **non-zero** normalized attention coefficients, incurring the revival of pruned connections during the neighborhood aggregation phase (hence, the corresponding edge is NOT actually pruned).
>
> Our experimental results using a revised implementation revealed a significant performance decrease compared to the results reported in [4]. However, we expect the performance enhancement of TEDDY as well as baselines if we adhere to the original implementation of UGS, since in this case, the amount of information received from neighbors also increases for our method. The same issue is pertinent on WD-GLT [5] as well, and we posit that this is due to the implementation on top of the UGS. Besides, we detailed the additional reproducing issue of WD-GLT in Appendix B.3.
>
> ---
> **References**
>
> [1] Pruning graph neural networks by evaluating edge properties, Knowledge-Based Systems 2022.
>
> [2] Effective Sparsification of Neural Networks with Global Sparsity Constraint, CVPR 2021.
>
> [3] Adversarial structured neural network pruning, CIKM 2019.
>
> [4] A Unified Lottery Ticket Hypothesis for Graph Neural Networks, ICML 2021.
>
> [5] Rethinking Graph Lottery Tickets: Graph Sparsity Matters, ICLR 2023.

---

### Official Review · Reviewer_SQ4D · 2023-10-29

**Soundness:** 3 good
**Presentation:** 4 excellent
**Contribution:** 3 good
**Rating:** 8
**Confidence:** 2

**Summary:**

This paper observes the importance of low-degree edges in the graph and proposes a one-shot edge sparsification framework that leverages edge-degree to find graph lottery tickets (GLT). They achieve superior performances on diverse benchmark datasets.

**Strengths:**

The graph lottery tickets problem is an important direction, and the solution proposed in this paper is simple and elegant by utilizing the low-degree edges. The experimental results are also convincing.

**Weaknesses:**

Could the authors compare the consumed time of the proposed method with other baselines to further emphasize the efficiency?

**Questions:**

Could the authors compare the consumed time of the proposed method with other baselines to further emphasize the efficiency?

---

> ### Author Response · Authors · 2023-11-21
> **Response to Reviewer SQ4D**
>
> We sincerely appreciate the reviewer’s helpful and constructive feedbacks. We would like to address the reviewer’s concern as below.
>
> **[On Wall-clock Time]**
>
> **Q1.** In response to the the reviewer’s valuable recommendation, we compared the wall-clock time analysis of TEDDY against baselines, averaged over five runs in five distinct simulations. We conducted experiments in the same GPU and GPU models for baselines and our method for each setting, to ensure fair comparisons.
>
> Overall, the results clearly substantiates that TEDDY consistently achieves a significant reduction in pruning time relative to the baselines. This is especially pronounced in GIN on the Pubmed dataset, where TEDDY outperforms WD-GLT [1] by a remarkable margin, displaying a maximum time saving of **220.72** seconds during the 15-th simulation. Similar trend is observed in large-scale datasets, where TEDDY’s time consumption is nearly half that of UGS [2]. The maximum duration gap is revealed in the Reddit dataset, with TEDDY concluding the last simulation **131.84** seconds quicker than UGS in GCN. For your convenience, we attached tables below in the revised manuscript, throughout Table 3-5.
>
> ---
> **References**
>
> [1] Rethinking Graph Lottery Tickets: Graph Sparsity Matters, ICLR 2023.
>
> [2] A Unified Lottery Ticket Hypothesis for Graph Neural Networks, ICML 2021.

---

> > ### Author Response · Authors · 2023-11-21
> > **Response to Reviewer SQ4D (Experiments on Small- and Medium-scale Datasets)**
> >
> > |  Datasets & GNNs     |  Methods      | 1-st               | 5-th               | 10-th              | 15-th              | 20-th              |
> > |-------|--------|---------------------|---------------------|---------------------|---------------------|---------------------|
> > | **Cora** |        |                     |                     |                     |                     |                     |
> > | GCN   | UGS    | 6.30 ± 0.55        | 5.84 ± 0.39        | 5.57 ± 0.80        | 5.65 ± 0.59        | 5.56 ± 0.73        |
> > |       | WD-GLT | 54.11 ± 1.28       | 53.61 ± 0.39       | 53.59 ± 0.17       | 53.52 ± 0.23       | 53.77 ± 0.37       |
> > |       | **TEDDY**  | **2.12 ± 0.08**    | **2.15 ± 0.16**    | **2.09 ± 0.12**    | **2.11 ± 0.13**    | **2.18 ± 0.05**    |
> > | GAT   | UGS    | 7.58 ± 0.44        | 7.11 ± 0.51        | 6.97 ± 0.59        | 7.03 ± 0.60        | 7.26 ± 0.24        |
> > |       | WD-GLT | 83.81 ± 10.02      | 79.98 ± 8.09       | 80.76 ± 6.30       | 80.84 ± 6.29       | 79.40 ± 4.81       |
> > |       | **TEDDY**  | **3.00 ± 0.07**    | **2.96 ± 0.14**    | **3.04 ± 0.08**    | **2.93 ± 0.40**    | **3.01 ± 0.05**    |
> > | GIN   | UGS    | 4.38 ± 0.81        | 4.11 ± 0.29        | 4.11 ± 0.28        | 4.06 ± 0.30        | 4.01 ± 0.19        |
> > |       | WD-GLT | 68.55 ± 10.13      | 72.43 ± 10.91      | 70.76 ± 10.45      | 72.04 ± 10.23      | 72.67 ± 11.89      |
> > |       | **TEDDY**  | **1.86 ± 0.25**    | **1.91 ± 0.31**    | **1.85 ± 0.28**    | **1.86 ± 0.32**    | **1.86 ± 0.27**    |
> > | **Citeseer** |        |                     |                     |                     |                     |                     |
> > | GCN   | UGS    | 8.21 ± 1.72        | 7.44 ± 0.52        | 7.55 ± 0.53        | 7.36 ± 0.55        | 7.33 ± 0.42        |
> > |       | WD-GLT | 83.73 ± 16.75      | 78.24 ± 14.09      | 77.67 ± 19.85      | 70.75 ± 22.62      | 69.51 ± 17.12      |
> > |       | **TEDDY**  | **2.65 ± 0.33**    | **2.72 ± 0.29**    | **2.70 ± 0.34**    | **2.71 ± 0.29**    | **2.80 ± 0.34**    |
> > | GAT   | UGS    | 8.78 ± 0.84        | 8.09 ± 0.52        | 8.47 ± 0.36        | 8.64 ± 0.77        | 8.41 ± 0.64        |
> > |       | WD-GLT | 93.96 ± 20.92      | 86.23 ± 19.97      | 89.55 ± 22.03      | 89.03 ± 22.84      | 86.74 ± 21.02      |
> > |       | **TEDDY**  | **3.21 ± 0.57**    | **3.35 ± 0.46**    | **3.51 ± 0.50**    | **3.55 ± 0.49**    | **3.63 ± 0.51**    |
> > | GIN   | UGS    | 6.61 ± 1.11        | 6.44 ± 0.77        | 6.05 ± 0.38        | 5.98 ± 0.38        | 6.16 ± 0.42        |
> > |       | WD-GLT | 108.74 ± 3.78      | 109.59 ± 4.25      | 106.69 ± 3.24      | 109.91 ± 4.16      | 107.93 ± 3.35      |
> > |       | **TEDDY**  | **2.51 ± 0.10**    | **2.53 ± 0.11**    | **2.46 ± 0.09**    | **2.49 ± 0.05**    | **2.52 ± 0.16**    |
> > | **Pubmed** |        |                     |                     |                     |                     |                     |
> > | GCN   | UGS    | 8.40 ± 0.54        | 7.80 ± 0.31        | 7.61 ± 0.55        | 7.95 ± 0.27        | 8.12 ± 0.30        |
> > |       | WD-GLT | 176.48 ± 26.20     | 161.88 ± 31.51     | 160.96 ± 34.56     | 141.58 ± 9.83      | 142.68 ± 12.39     |
> > |       | **TEDDY**  | **3.23 ± 0.20**    | **3.27 ± 0.13**    | **3.16 ± 0.14**    | **3.14 ± 0.21**    | **3.08 ± 0.12**    |
> > | GAT   | UGS    | 9.85 ± 0.91        | 9.35 ± 0.59        | 9.16 ± 0.72        | 9.61 ± 0.67        | 9.41 ± 0.75        |
> > |       | WD-GLT | 69.64 ± 8.76       | 68.69 ± 8.31       | 66.07 ± 5.49       | 64.38 ± 2.74       | 63.66 ± 1.11       |
> > |       | **TEDDY**  | **3.90 ± 0.26**    | **3.82 ± 0.18**    | **3.58 ± 0.20**    | **3.63 ± 0.21**    | **3.50 ± 0.30**    |
> > | GIN   | UGS    | 7.42 ± 1.01        | 7.17 ± 0.41        | 7.11 ± 0.51        | 7.03 ± 0.49        | 6.95 ± 0.41        |
> > |       | WD-GLT | 218.87 ± 6.40      | 214.61 ± 7.44      | 213.63 ± 5.83      | 223.74 ± 4.99      | 216.23 ± 6.28      |
> > |       | **TEDDY**  | **3.20 ± 0.18**    | **3.12 ± 0.13**    | **3.06 ± 0.10**    | **3.02 ± 0.07**    | **2.97 ± 0.12**    |

---

> > > ### Author Response · Authors · 2023-11-21
> > > **Response to Reviewer SQ4D (Experiments on Large-scale Datasets)**
> > >
> > > |    Datasets & GNNs      |   Methods   | 1-st               | 5-th               | 10-th              | 15-th              | 20-th              |
> > > |----------|------|---------------------|---------------------|---------------------|---------------------|---------------------|
> > > | **Arxiv** |      |                     |                     |                     |                     |                     |
> > > | GCN      | UGS  | 82.28 ± 19.32       | 77.93 ± 13.14       | 75.26 ± 12.89       | 73.18 ± 11.94       | 71.50 ± 10.81       |
> > > |          | WD-GLT | N/A               | N/A                 | N/A                 | N/A                 | N/A                 |
> > > |          | **TEDDY** | **43.45 ± 0.02**   | **39.70 ± 0.04**    | **36.10 ± 0.05**    | **33.35 ± 0.03**    | **31.16 ± 0.02**    |
> > > | SAGE     | UGS  | 74.22 ± 3.98       | 73.37 ± 2.61        | 76.18 ± 4.86        | 75.18 ± 5.07        | 70.94 ± 1.00        |
> > > |          | WD-GLT | N/A               | N/A                 | N/A                 | N/A                 | N/A                 |
> > > |          | **TEDDY** | **45.10 ± 0.02**   | **41.96 ± 0.02**    | **38.85 ± 0.02**    | **36.43 ± 0.01**    | **34.58 ± 0.02**    |
> > > | **Reddit** |    |                     |                     |                     |                     |                     |
> > > | GCN      | UGS  | 180.80 ± 69.56     | 176.25 ± 54.61      | 158.97 ± 52.54      | 154.12 ± 42.40      | 155.35 ± 49.46      |
> > > |          | WD-GLT | OOM              | OOM                 | OOM                 | OOM                 | OOM                 |
> > > |          | **TEDDY** | **47.68 ± 0.02**   | **40.29 ± 0.03**    | **32.92 ± 0.02**    | **27.56 ± 0.01**    | **23.51 ± 0.10**    |
> > > | SAGE     | UGS  | 174.47 ± 34.80     | 173.08 ± 31.65      | 167.41 ± 32.09      | 166.55 ± 30.85      | 161.88 ± 30.77      |
> > > |          | WD-GLT | OOM              | OOM                 | OOM                 | OOM                 | OOM                 |
> > > |          | **TEDDY** | **65.53 ± 0.03**   | **55.77 ± 0.03**    | **46.25 ± 0.03**    | **39.12 ± 0.05**    | **33.45 ± 0.04**    |

---

### Official Review · Reviewer_mYP6 · 2023-10-30

**Soundness:** 3 good
**Presentation:** 3 good
**Contribution:** 3 good
**Rating:** 6
**Confidence:** 3

**Summary:**

The paper proposes TEDDY, a novel edge sparsification framework that considers the structural information of the graph. TEDDY sparsifies graph edges based on scores designed by utilizing edge degrees, and sparsifies parameters via projected gradient descent on $l_0$ ball. In particular, sparsification of both edges and parameters can be done in one-shot and the edge sparsification part does not to consider node features. Then the paper demonstrates the effectiveness of TEDDY over iterative GLT methods with comprehensive experiments.

**Strengths:**

1.	The paper is overall easy to follow.
2.	The experiments are diverse and thorough.
3.	It is an interesting observation that low-degree edges are important, which is supported both empirically and theoretically. The proposed method is thus principally designed.
4.	The proposed method is one-shot and efficient (but seems still need to train a dense network first; see weaknesses).
5.	The proposed method does more than just preserving the performance of vanilla dense GNNs---it actually improves the original performance in many settings. This is an impressive and interesting result.

**Weaknesses:**

1.	The main part of the proposed method and some discussion of the edge degrees are not clear and confusing.
- It is not very clear to me how T in eq.(5) is actually used (just dropping the edges with the lowest scores)? Also it seems that T computes all scores for all node pairs. How to deal the case when there is no edge between two nodes?
- Typo in eq.(5)? Should it not include specific node v instead?
- The analysis of the effect of edge degree in Section 4 only applies for first-order degrees, yet the paper discusses why higher-order edge degrees need to be considered heuristically via a toy example and claims TEDDY is based on “multi-level consideration of degree information“ in section 5.1. Then in the experiments, it never touches upon higher-order edge degree information and only discusses about first-order edge degrees in Figure 8.
2.	The loss objective involves distillation (6), meaning that one still needs to train a dense network on the entire graph first, weakening the efficiency of the proposed method.

**Questions:**

1. Have you evaluated the zero-shot performance of the proposed edge sparsification method (with dense weights)? Intuitively it seems that it might able to work in the zero-shot setting, and in this way, one can single out the effect of sparse subgraph on the boost of performance.
2.	What happens if the distillation term in (6) is removed (so one does not to try a dense network at first)? How significantly the performance would be affected? Could you provide an ablation study on this?
3.	Would TEDDY work if one trains on small graphs and then applies to large graphs?
4.	Any intuition why TEDDY is still able to improve the performance of vanilla GNNs under extreme sparsity?

---

> ### Author Response · Authors · 2023-11-21
> **Response to Reviewer mYP6 (1/2)**
>
> We sincerely appreciate the reviewer’s helpful and constructive feedbacks. We would like to address the reviewer’s concerns as below.
>
> **[Computation of Edge Score]**
>
> **W1-1.** $T_{edge}\in\mathbb{R}^{|\mathcal{E}|}$ is only computed ***for node pairs that exist as edges***, not for all possible node pairs. In practice, this is achieved by element-wise multiplication between $\widetilde g(v)$ of nodes $v$ in rows (i.e., source nodes) and $\widetilde g(u)$ of nodes $u$ (i.e., destination nodes) in columns of the edge index provided in Pytorch Geometric [1]. This ensures the obtainment of scores for only real edges. To clarify this, we have attached the code snippet of the edge score calculation in supplementary material (`compute_edge_score` function in `pruning.py`).
>
> ---
> **[Typo in Eq. (5)]**
>
> **W1-2.** Thank you for the reviewer’s correction. We modified the typo in Eq. (5) in the revised paper.
>
> ---
> **[On Degree Analysis]**
>
> **W1-3.** We apologize to the reviewer for confusion. We provide the clarification as follows. We empirically validated the necessity to consider higher-order edge degrees in Figure 17 in the appendix. As evidenced by the figure, the multi-level consideration of edge degree clearly improves the performance (specified as Ours), compared to the outcomes of pruning based solely on first-order degree information (specified as 1-hop Degree). While Figure 8 focuses on first-order degrees, the motivation behind Figure 8 is on the particular emphasis of TEDDY performing degree-aware pruning, reflecting the prior observation highlighted in Section 4. Conversely, baselines lacks the edge degree consideration, correlating with their less optimal performance as compared to our method.
>
> ---
> **[On Distillation]**
>
> **W2.** Our findings, as illustrated in Figure 17 and 18, do indicate performance improvement due to the model distillation. Note that, however, our analysis puts more emphasis on the significance of 2-hop degree based edge score, $T_{edge}$, in Figure 18 in revised paper. As demonstrated in the figure, the design of edge scores is more critical, since any configuration with the baseline including distillation fails to achieve better pruning performance than our method with a sole $T_{edge}$ (represented as Ours w/o $\mathcal L_{dt}$).
>
> ---
> **[On Zero-shot Performance]**
>
> **Q1.** In the development of TEDDY, we specifically focused on a one-shot edge sparsification approach, which inherently differs from zero-shot pruning strategies. Consequently, evaluating TEDDY in a zero-shot setting falls outside the scope of our current research. This distinction aligns with the methodology used in previous studies on GNN pruning [2, 3], which typically employ pretrained models as a foundation for iterative pruning. While zero-shot sparsification offers an interesting perspective, our method gains significant efficacy on one-shot mechanism, by eliminating the need for repeated retraining each pruning level adopted in [2, 3] (Please refer to the wall-clock time analysis of TEDDY against baselines in Appendix A.8 in the revised paper).
>
> ---
> **[On Distillation Component Removal]**
>
> **Q2.** In response, we present the ablation study in Figure 17. As illustrated in the figure, removing distillation term in TEDDY (specified as Ours w/o $\mathcal L_{dt}$) consistently achieves superior performance compared to baselines, even with those integrated with distillation component.
>
> Further ablation on all benchmark datasets is detailed in Figure 18 in the revised paper. In the figure, the performance UGS [2] attached with distillation loss remains suboptimal compared to TEDDY without $\mathcal L_{dt}$, regardless of benchmark datasets. This underscores the significant influence of our degree-based pruning component. Additional studies to further evaluate this effect on other architectures will be included in the later revision.

---

> ### Author Response · Authors · 2023-11-21
> **Response to Reviewer mYP6 (2/2)**
>
> **[On Smaller Graphs to Larger Graphs]**
>
> **Q3.** The reviewer's question raises a consideration about the scalability of TEDDY, which may be interpreted as its adaptability to an inductive semi-supervised node classification setting, where the fraction of the validation and test set is unobserved and larger than that of the training set. In this perspective, GNNs are trained on the partial set of the entire dataset while tested on the remaining unseen portions.
>
> In response to the reviewer’s insightful question, we expanded our experiments to assess our model’s performance in inductive scenarios. We adjusted the Cora, Citeseer, and Pubmed datasets to a 20/40/40% split for training, validation, and testing, guaranteeing no exposure to the validation and test nodes as well as edges connected to them during training.
>
> In the corresponding results, illustrated in Figure 15 and 16 of the revised manuscript, TEDDY achieves prominent pruning performance on inductive setting as well. This is especially pronounced on the Cora dataset with GCN and on the Citeseer dataset across all benchmark architectures, surpassing the accuracy of vanilla GNNs. Hence, the results affirm TEDDY’s capacity to generalize from smaller to larger graphs effectively.
>
> ---
> **[On Extreme Sparsity]**
>
> **Q4.** Following the reviewer’s insightful question, we believe that TEDDY strategically preserves low-degree nodes while pruning primarily focuses on high-degree nodes. This approach is rooted in the inherent design of GNNs, where neighborhood aggregation is a core component. Unlike vision domain, which often relies on complex, fully connected layers, GNNs typically employ lighter pre-aggregation embeddings. This characteristic is particularly crucial in such neural networks, where low-degree nodes have fewer neighbors to gather information from. Excessive pruning that drastically reduces the degree of these nodes, or in extreme cases, isolates them (equivalent to the forward pass of MLP), can severely hamper the model's ability to classify, since the amount of information they can obtain through the message-passing scheme becomes limited. By protecting these low-degree nodes, TEDDY ensures that even under extreme sparsity, there is sufficient local structure to facilitate effective classification, hence improving the performance of vanilla GNNs.
>
> ---
> **References**
>
> [1] Fast graph representation learning with PyTorch Geometric, ICLR 2019 (RLGM Workshop).
>
> [2] A Unified Lottery Ticket Hypothesis for Graph Neural Networks, ICML 2021.
>
> [3] Rethinking Graph Lottery Tickets: Graph Sparsity Matters, ICLR 2023.

---

### Official Review · Reviewer_vxbf · 2023-10-30

**Soundness:** 2 fair
**Presentation:** 3 good
**Contribution:** 3 good
**Rating:** 6
**Confidence:** 3

**Summary:**

This paper proposes a one shot graph pruning algorithm to find Graph Lotttery Tickets (GLTs) by (i) deleting graph edges from nodes with higher degrees and (ii) sparsifying node parameters using $l_0$ regularization with Projected Gradient Descent.
The sparse networks obtained by the authors are shown to improve performance over existing Graph Lottery Ticket methods.

**Strengths:**

1. The proposed method finds GLTs using a one shot training approach outperforming existing iterative graph pruning algorithms.

**Weaknesses:**

1. While the proposed idea of sparsifying the graph edges based on the degree information is simple and makes intuitive sense (as also shown empirically), why do the authors consider only the degree information as opposed to other metrics like spectral information of the graph or centrality which might convey more information about the graph. Is the degree information sufficient in this regard to obtain a sufficiently sparse graph in comparison with these other metrics? Recent work has shown randomly dropping graph edges can also help training, in comparison to the proposed idea in Yu et al.  [1], how does randomly pruning graph edges compare?

2. The parameters sparsification method used is PGD with $l_0$ regularization. But there is no justification provided for using this particular method. There are other continuous sparsification schemes that have shown to achieve highly sparse networks in a single training run for feedforward networks like Kusupati et al. [2] and Louizos et al. [3].

3. The authors use GATs and GCNs in their experiments. However, the structure of GATs allows them to inherit some degree information in the form of attention while GCNs do not. Does this difference change the performance of the proposed graph pruning criterion for either architectures?

4. The results in Table 1 might be better visualized via a heatmap showing how much Graph sparsity and Weight sparsity can be achieved and if there is a tradeoff between the two.


[1] Rong, Yu, et al. "Dropedge: Towards deep graph convolutional networks on node classification." *arXiv preprint arXiv:1907.10903* (2019).

[2] Kusupati, Aditya, et al. "Soft threshold weight reparameterization for learnable sparsity." *International Conference on Machine Learning* 2020.

[3] Louizos, Christos, Max Welling, and Diederik P. Kingma. "Learning Sparse Neural Networks through L_0 Regularization." *International Conference on Learning Representations*. 2018.

**Questions:**

1. The proposed method uses a pretraining strategy on the full graph before the pruning step. Are the parameters reinitialized at the end of the pruning stage like lottery tickets or does training continue after graph sparsification?

2. The authors mention the use of multilevel degree information for the graph pruning criterion in Eq. 4. However, it is not clear if the number of hops considered is determined by the number or layers in the network or is a hyperparameter that is tuned?

---

> ### Author Response · Authors · 2023-11-21
> **Response to Reviewer vxbf (1/2)**
>
> We sincerely appreciate the reviewer’s helpful and constructive feedbacks. We would like to address the reviewer’s concerns as below.
>
> **[On Motivation]**
>
> **W1-1.** While we agree with the reviewer that considering aspects such as spectral information or centrality is an appealing approach, it is worth noting that in case of large graphs, such as Reddit or ArXiv dataset, understanding the spectral properties of the graph Laplacian/adjacency matrix is often computationally infeasible. To consider spectral information, one would require information for spectrums (e.g. eigenvectors/eigenvalues), and exploiting such information necessitates algorithms like eigendecomposition, which requires a expensive time complexity of $O(|V|^3)$. Thus, our primary goal is to design a pruning algorithm that can quickly identify to-be-pruned edges even for large graphs. As an initial step toward our goal, we focus on considering degree information and believe that we successfully demonstrated that even with only degree information, our TEDDY could effectively maintain the performance of dense graphs/GNNs through extensive experiments (Section 6). However, in line with the reviewer's suggestion, we acknowledge the potential to incorporate a more diverse range of graph information and consider it as our future avenue.
>
> ---
> **[On Comparison with Random Edge Pruning]**
>
> **W1-2.** In response to the reviewer’s insightful suggestion, we expanded our experiments including random dropping strategy in DropEdge [1]. In this experiment, we solely conducted a graph sparsification to directly compare the impacts of diverse edge pruning approaches, including our TEDDY.
>
> According to the results illustrated in Figure 19, TEDDY consistently surpasses baseline approaches including DropEdge. Notably, our method not only preserves but outperforms the accuracy of vanilla GNNs in the Citeseer and Pubmed datasets across all benchmark GNNs. Figure 19 also demonstrates that utilizing DropEdge technique results in suboptimal performance, underscoring the necessity of sophisticated pruning strategy. Interestingly, in a distinct graph sparsification scenario, UGS [2] tends to exhibit the similar performance with that of the DropEdge.
>
> ---
> **[On Parameter Sparsification]**
>
> **W2.** The most crucial aspect of our PGD on $\ell_0$-ball is its capability **for one-shot pruning at any given sparsity level**. For example, the previous approaches like UGS and WD-GLT require sensitive tuning of the $\ell_1$ regularization parameter due to an iterative process to reach the target sparsity level, while our method enables one-shot pruning without any regularization parameter. In a similar sense, references [3], [4], as the reviewer brings to us, also necessitate tuning of the regularization parameter to achieve the target sparsity level, which may be burdensome in practice. Additionally, by encouraging sparsity during training, we can gain computational advantages in terms of actual number of operations even during the training phase.
>
> ---
> **[On Architectures]**
>
> **W3.** In addressing the differences between GATs and GCNs in our experiments, it's important to note the issue of the public code of UGS [2] on GAT (https://github.com/VITA-Group/Unified-LTH-GNN/blob/main/NodeClassification/gnns/gat_layer.py). According to the code, edge masking is applied *prior to* the softmax normalization of attention coefficients. However, this implementation leads to a notable issue where pruned edges inevitably receive **non-zero** normalized attention coefficients, incurring the revival of pruned connections during the neighborhood aggregation phase. Our experimental results using a revised implementation revealed a significant performance decrease compared to the results reported in [2]. This suggests that the removal of edges *before forwarding GAT layer*, may not significantly impact the performance if the attention mechanism inherently captures degree information beneficial for pruning. Given these observations, it remains unclear whether the attention mechanism in GATs offers a substantial advantage over GCN in encoding degree information.

---

> ### Author Response · Authors · 2023-11-21
> **Response to Reviewer vxbf (2/2)**
>
> **[On Table 1]**
>
> **W4.** We appreciate the reviewer’s helpful feedback. In response to the reviewer’s suggestion, we present the results of our method on GIN under extreme sparsity as a heatmap in Figure 20, in the revised paper. The performance in each heatmap element is an average performance over 5 random seeds.
>
> Overall, the performance of TEDDY is not uniformly affected by increased sparsity. In the Cora dataset, our method demonstrates resilience to higher graph sparsity levels, surpassing the vanilla GIN’s accuracy of 76.34%. This suggests that a sparser graph generated from our method has a potential to mitigate overfitting and achieve generalization. For the Citeseer dataset, the performance remains relatively stable across a range of weight sparsity levels, underscoring a robustness to parameter reduction. Furthermore, all configurations exceed the vanilla GIN's accuracy of 68.1%, strongly indicating the effectiveness of TEDDY across diverse sparsities. The similar efficacy is observed in the Pubmed dataset, where all elements surpasses the vanilla GIN’s performance of 77.9%. Interestingly, we observe a trend in the Pubmed where the performance generally improves with increased sparsity, with the highest accuracy achieved under the most extreme sparsity ($p_g=p_\theta=85$%).
>
> ---
> **[On parameter reinitialization at the end of the pruning stage]**
>
> **Q1.** Our method does not adhere to the standard lottery ticket hypothesis framework, where parameters are reinitialized to their original values during the post-pruning. Instead, our approach is **agnostic** to the initial parameter values. Following the graph sparsification, we do not revert to the original initialization but continue training from the current initialized state of the parameters, which may be randomly initialized or pre-trained. Hence, TEDDY can obtain robust performance as the original GNNs, irrespective of the initial starting points of the parameters.
>
> ---
> **[The number of hops considered for multi-level degree consideration]**
>
> **Q2.** In our method, as outlined in Eq. 3-5, we considered two-hop degree information to calculate edge-wise scores $T_{edge}$ for all settings. There could be other options for the choice of the number of hops, but we fixed it as two, since two hop degree consideration is sufficient to yield prominent performance. Consequently, the number of hops in our pruning criterion is a predetermined choice.
>
> ---
> **References**
>
> [1] Dropedge: Towards deep graph convolutional networks on node classification, ICLR 2020.
>
> [2] A Unified Lottery Ticket Hypothesis for Graph Neural Networks, ICML 2021.
>
> [3] Soft threshold weight reparameterization for learnable sparsity, ICML 2020.
>
> [4] Learning Sparse Neural Networks through L_0 Regularization, ICLR 2018.

---

> > ### Comment · Reviewer_vxbf · 2023-11-22
> > **Response to Rebuttal**
> >
> > I thank the author's for additional experiments and revising the manuscript. I am satisfied with their response and I will keep my score for acceptance.
> >
> > I wanted to follow up on the W3 above, in the case where the input to the attention is zero the output of the softmax would be nonzero. However, if the edge has been masked out, it should not be included in the aggregation, which would imply that the attention value does not matter?

---

> > > ### Author Response · Authors · 2023-11-22
> > > **Response to Reviewer vxbf**
> > >
> > > We sincerely thank the reviewer vxbf for the reassessment and positive opinion for our work. Your constructive feedback has enhanced our work during the rebuttal. We are glad that our revision and responses have addressed your concerns.
> > >
> > > In the original code of the UGS [1], the edge mask is only applied to the edge weight, i.e., *pre-normalized* attention coefficient, **NOT the embeddings of nodes**. Hence, since the softmax-normalized attention coefficient for a pruned edge $(i, j)$ will be non-zero according to the original code, the aggregation for the update of node $i$ will involve the **embedding of node $j$**.
> > >
> > > ---
> > > **References**
> > >
> > > [1] A Unified Lottery Ticket Hypothesis for Graph Neural Networks, ICML 2021.

---

### Official Review · Reviewer_k6Ge · 2023-10-31

**Soundness:** 2 fair
**Presentation:** 2 fair
**Contribution:** 2 fair
**Rating:** 5
**Confidence:** 4

**Summary:**

This paper introduces several techniques to sparsify graph data and networks in order to build a sparse graph learning model. The important aspects are as follows:
1) Degree-based graph sparsification to remove low-degree edges and make graphs sparse;
2) Distillation by matching the classification logits of the model pre-trained on the whole graphs;
3) Parameter sparsification by using a subset of parameters (i.e., Projected Gradient Descent).

When applied to classical GNNs such as GCN/GAT/GIN, the experimental results validate the proposed method by showing some performance improvements.

**Strengths:**

The proposed methods are generally straightforward and concise. They empirically improve performance compared to the baselines. The whole process is clearly stated in Algorithm 1.

**Weaknesses:**

1. Baseline GNN models are somewhat outdated. Traditional GNNs such as GCN/GAT/GIN are known to suffer from limited expressivity, resulting in restricted empirical performances in many cases. Several studies have done expressiveness analysis, to name a few references [3,4]. Later works, such as graph transformers and expressive GNNs [1,2,3], have proven to be more powerful and meaningful for empirical studies.

2. Detailed ablation experiments are lacking, which are necessary to clearly validate the effectiveness of each component.

3. The theoretical analysis concerning the motivation behind preserving low-degree edges seems somewhat disconnected from the main idea.

4. Some claims regarding training efficiency and generalization performance appear to be misleading.

These points are detailed in the questions section below.



[1] Recipe for a general, powerful, scalable graph transformer. NeurIPS 2022.\
[2] Specformer: Spectral Graph Neural Networks Meet Transformers. ICLR 2023.\
[3] Provably Powerful Graph Networks. NeurIPS 2019.\
[4] Weisfeiler and Leman Go Neural: Higher-order Graph Neural Networks. AAAI 2019.

**Questions:**

1. Conventional GNNs are empirically seen to suffer from limited expressivity. How do the latest transformer-based GNNs or more expressive models like GPS[1], Specformer[2], and PPGN[3] compare when applying the proposed techniques? These newer models have already surpassed the empirical performance of older baselines like GCN/GAT/GIN. Validating the proposed techniques on these latest methods rather than older models would make the arguments more convincing.

2. About Figure 1: What is the precise step for removing high-degree or low-degree edges? Given a graph sparsity value, how exactly is the subgraph generated? Is random sampling involved in this process? If so, could you provide the results of multiple runs (e.g., mean and standard deviation) to separate the training-related noise?

3. Ablation Studies: Could you provide detailed ablation studies on the effectiveness of the proposed three components? a) graph sparsification by removing edges; b) model distillation; c) parameter sparsification with projected gradient descent. It seems that these components could be applied separately to any of the baseline training methods (UGS/WD-GLT) in empirical studies.
In Figure 13, there are some comparisons made. Graph sparsification doesn’t seem to improve empirical performance at all. There are no curves representing UGS/WD-GLT without distillation loss. Although considering multi-hop subgraphs appears to be effective in a way, this concept is not new in graph learning literature (e.g., k-WL GNNs [4]).

4. The theoretical analysis involving the upper bound of the symmetrically normalized adjacency matrix is somewhat unclear.
a) Concerning the mentioned analysis, how realistic is the assumption of the Lipschitz constant being present?
b) If the aforementioned assumptions hold true for the specified models, how plausible is it that removing low-degree edges actually increases the term $\\frac{\\text{deg}\_\text{max} + 1}{\\text{deg}\_\text{min} + 1}$? This aspect may be specific to datasets and could possibly be numerically simulated.
c) Conversely, removing high-degree edges appears to reduce the generalization gap by lowering $\\text{deg}\_\text{max}$, according to the preceding analysis. What is the reason this aspect is not emphasized or motivated more within the discussion?

4. In Eq. (5), the edge score is always symmetric. How does this approach apply to directed graphs?

5. How crucial is the distillation training? It appears to undermine the purpose of utilizing sparse graphs, as there is a necessity to initially train the model on the full graphs. This approach also seems to contradict the claim of "a single training," a statement that is reiterated numerous times throughout the paper.

6. The paper asserts that "TEDDY significantly surpasses conventional iterative approaches in generalization." Could you provide more details to substantiate this claim? Which sections or aspects of the experiments corroborate this assertion regarding generalization?

7. The paper claims that the employed PGD training saves computation time compared to the iterative approach. Could you provide additional results to validate this claim, such as comparing training hours using the same GPU hardware?

[1] Recipe for a general, powerful, scalable graph transformer. NeurIPS 2022.\
[2] Specformer: Spectral Graph Neural Networks Meet Transformers. ICLR 2023.\
[3] Provably Powerful Graph Networks. NeurIPS 2019.\
[4] Weisfeiler and Leman Go Neural: Higher-order Graph Neural Networks. AAAI 2019.

---

> ### Author Response · Authors · 2023-11-21
> **Response to Reviewer k6Ge (1/8)**
>
> We sincerely appreciate the reviewer’s helpful and constructive feedbacks. We would like to address the reviewer’s concerns as below.
>
> **[On Experiments for Recent GNNs]**
>
> **Q1.** In response to the reviewer’s insightful suggestion, we expanded our experiments to encompass recent Transformer-based GNN architectures. While integrating TEDDY with these advanced models, we encountered several issues with some architectures suggested by the reviewer.
>
> To begin with, PGNN [1] utilizes a **dense** adjacency matrix in the input tensor, resulting in an $N\times N$ matrix for a graph with $N$ nodes. In general, adjacency matrix in GNNs are encoded as a **sparse** format, where the computation can be highly dependent on the number of edges. This is where the graph sparsification can offer a benefit. However, in the case of dense matrix representations as PGNN, both zero and non-zero elements are treated equally in terms of storage and computational overhead. Consequently, the inherent characteristics of PGNN's dense matrix approach do not align well with the advantages offered by graph sparsification.
>
> Regarding GraphGPS [2], its primary efficacy is demonstrated in graph-level tasks, diverging from our focus on node classification. Similarly, Specformer [3], while benchmarked against GraphGPS in graph-level tasks, was evaluated against other baselines in node-level tasks. In our experiments, we observed that GraphGPS did not yield satisfactory performance on Cora and Citeseer compared to classic GNNs.
>
> Consequently, we explore alternative graph transformers, specifically UniMP [4], NAGphormer [5], and Specformer [3], which have demonstrated strong performance in node classification without significant degradation on our benchmark datasets. However, both Specformer and NAGphormer replace the input adjacency matrix with eigenvectors and eigenvalues from eigendecomposition. The baselines’ differentiable mask for the adjacency matrix requires backpropagation through eigenvectors and eigenvalues, introducing a practical challenge with high complexity of $O(N^3)$ for the number of node $N$ per iteration. Given these constraints, we conducted the evaluation of these two transformer architectures only on TEDDY, whereas including experiments on TEDDY and all baselines for UniMP [4], which has also been recognized as a graph transformer benchmark in [6].
>
> As depicted in Figure 13 and 14, TEDDY accomplishes stable pruning performance, surpassing all baselines when equipped with UniMP as a backbone. In particular, UGS and WD-GLT show significant degradation as the sparsity increases, whereas the performance of TEDDY is stable across all benchmark datasets. Note that differentiable mask approach from baselines may prune edges more than pre-defined ratio, due to the possibility of multiple mask elements having the same value. Our method also yields decent performance in NAGphormer and Specformer, notably in NAGphormer on the Pubmed dataset, with a marginal accuracy loss of 0.56. Overall, these results demonstrate TEDDY's versatility across diverse foundational architectures.

---

> > ### Comment · Reviewer_k6Ge · 2023-11-22
> >
> > Thank you for your feedback. Regarding Figures 13 and 14, there are just minor variations in the y-axis of the performance curves. In similar studies, such as UniMP [4], NAGphormer [5], and Specformer [3], the standard deviation is typically reported, providing a clearer understanding of performance variability. The slight differences observed at different graph sparsity in Figures 13 and 14 could potentially be explained by training noise. Could you offer additional insights on this matter?

---

> ### Author Response · Authors · 2023-11-21
> **Response to Reviewer k6Ge (2/8)**
>
> **[On Figure 1]**
>
> **Q2.** In the analysis of Figure 1, we define the edge score of each edge $(i,j)$ as $T_{edge}(i, j) =(|\mathcal N(i)|+|\mathcal N(j)|)/{2}$, where $\mathcal N(k)$ denotes the set of neighboring nodes connected to node $k$. For our experiments, we conduct 20 simulations where GNNs are trained after pruning edges based on their scores. Specifically, given a graph sparsity ratio $p_g$ for each simulation, we pruned $p_g\%$ of edges having the highest edge score $T_{edge}$ in the high degree pruning scenario (represented as blue lines in the figure), and pruned edges with the lowest scores in the low-degree scenario (represented as orange lines in the figure). Hence, our TEDDY does not involve any stochastic procedures such as random sampling.
>
> However, to address the reviewer’s suggestion, we conduct additional simulations with 5 different random seeds (0 to 4) and reported the average performance in Table 2. In the table, **deg_low** refers to pruning based on the lowest degree, while **deg_high** corresponds to the highest degree-based pruning. The results clearly demonstrate that pruning low-degree edges, i.e., low $T_{edge}$, can significantly deteriorate the performance, especially in GAT on the Pubmed dataset, where the performance between **deg_low** and **deg_high** reaches the maximum gap of 24.1% in the final simulation. For convenience, we attach the below tables in Table 2 in the revised manuscript.
>
> | Simulations |        | 1-st | 1-st  |  5-th | 5-th  | 10-th| 10-th | 15-th | 15-th | 20-th | 20-th |
> |-------------|--------|--------------|---------------|--------------|---------------|---------------|----------------|---------------|----------------|---------------|----------------|
> | GNNs        | Dataset|       deg_low       |      deg_high         |       deg_low       |          deg_high     |       deg_low        |          deg_high      |      deg_low         |         deg_high       |     deg_low          |       deg_high         |
> | GCN         | Cora   | 80.38 ± 0.19 | **81.28 ± 0.64** | 77.66 ± 0.41 | **77.92 ± 0.34** | 75.36 ± 0.42 | **77.68 ± 0.32** | 74.24 ± 0.16 | **76.02 ± 0.43** | 69.54 ± 0.32 | **72.62 ± 0.32** |
> |             | Citeseer | 70.44 ± 0.21 | **70.88 ± 0.28** | 67.0 ± 0.27 | **70.60 ± 0.18** | 64.78 ± 0.31 | **67.58 ± 0.72** | 63.46 ± 0.42 | **67.72 ± 0.37** | 63.22 ± 0.51 | **65.70 ± 0.56** |
> |             | Pubmed | 78.32 ± 0.19 | **79.18 ± 0.12** | 76.66 ± 0.79 | **77.66 ± 0.27** | **77.52 ± 0.28** | 77.32 ± 0.20 | **77.72 ± 0.17** | 76.74 ± 0.10 | 75.20 ± 0.20 | **76.54 ± 0.08** |
> | GAT         | Cora   | 77.64 ± 1.33 | **79.72 ± 1.06** | 68.34 ± 1.26 | **78.50 ± 1.16** | 56.72 ± 0.48 | **72.44 ± 1.10** | 50.12 ± 0.93 | **67.74 ± 0.92** | 44.78 ± 0.80 | **51.64 ± 1.67** |
> |             | Citeseer | 69.48 ± 1.36 | **70.96 ± 0.75** | 57.02 ± 0.64 | **66.86 ± 0.63** | 42.68 ± 2.14 | **59.74 ± 1.64** | 35.40 ± 1.50 | **49.86 ± 1.82** | 28.40 ± 0.97 | **43.24 ± 1.43** |
> |             | Pubmed | **78.40 ± 0.61** | 78.14 ± 1.01 | 71.66 ± 0.59 | **76.32 ± 0.55** | 58.48 ± 0.85 | **74.52 ± 0.56** | 49.52 ± 0.64 | **72.12 ± 0.64** | 44.10 ± 0.57 | **68.20 ± 0.61** |
> | GIN         | Cora   | 74.86 ± 1.14 | **76.48 ± 1.66** | 70.70 ± 1.13 | **76.84 ± 1.81** | 66.70 ± 0.33 | **74.70 ± 2.32** | 62.72 ± 1.38 | **72.96 ± 0.55** | 58.82 ± 1.99 | **70.28 ± 0.99** |
> |             | Citeseer | 67.52 ± 0.99 | **68.78 ± 0.98** | 63.80 ± 1.48 | **69.16 ± 0.96** | 62.32 ± 1.14 | **68.46 ± 0.48** | 60.60 ± 1.27 | **66.92 ± 0.27** | 59.48 ± 2.25 | **65.64 ± 1.09** |
> |             | Pubmed | 75.64 ± 0.72 | **77.96 ± 0.36** | 75.12 ± 0.66 | **77.38 ± 0.27** | 73.68 ± 1.26 | **77.44 ± 0.10** | 70.74 ± 1.39 | **74.88 ± 0.62** | 70.88 ± 2.72 | **73.30 ± 0.11** |

---

> ### Author Response · Authors · 2023-11-21
> **Response to Reviewer k6Ge (3/8)**
>
> **[On Ablation Study]**
>
> **Q3-1-a. (On graph sparsification)** We sincerely appreciate the reviewer’s constructive feedback. However, we would like to clarify that the purpose of graph sparsification is to **preserve the original GNN’s accuracy**, rather than improving it as a form of regularization. Hence, we respectfully disagree in that our graph pruning method doesn’t seem to improve empirical performance at all. And in fact, according to a sole graph sparsification performance comparison between our pruning approach and baselines, illustrated in Figure 19, our approach consistently achieves superior performance to baselines. Thus, this clearly substantiates that our graph sparsification indeed well-preserves, and in most cases even surpasses (**7 out of 9 settings** in Figure 19) the original GNNs’ performance.
>
> Furthermore, following the reviewer’s thoughtful opinion that components TEDDY can be applied separately, we provide comprehensive ablation studies for diverse combinations of our components. Utilizing GAT as a backbone architecture, we evaluated **7 distinct scenarios** detailed in Figure 18:
>
> (Scenario 1) TEDDY with all components (specified as Ours),
>
> (Scenario 2) TEDDY without distillation loss (specified as Ours w/o $\mathcal L_{dt}$),
>
> (Scenario 3) TEDDY with magnitude-based weight mask pruning (specified as Ours w/ $m_\theta$),
>
> (Scenario 4) TEDDY without weight sparsification, i.e., our graph sparsification method with dense weights (specified as Ours w/o weight spar.)
>
> (Scenario 5) UGS [7] with all components (specified as UGS),
>
> (Scenario 6) UGS with distillation loss (specified as UGS w/ $\mathcal L_{dt}$),
>
> (Scenario 7) UGS with $\ell_0$-based projected gradient descent (specified as UGS w/ $\ell_0$ PGD).
>
> ---
>
> **Q3-1-b. (On distillation loss)** The results, as depicted, do indicate performance improvement due to the model distillation. However, the results also clearly demonstrate that even when UGS is augmented with or without $\mathcal L_{dt}$ (scenario 6 and 5), it fails to match the performance of our framework **without** $\mathcal L_{dt}$ (scenario 2). Similar results regarding WD-GLT [8] are illustrated in Figure 17. This demonstrates that our edge sparsification method rooted in low edge degree possesses more significant impact than the distillation component in GNN pruning.
>
> ---
>
> **Q3-1-c. (On weight sparsification)** Furthermore, our method including $\ell_0$ PGD (scenario 1) shows comparable performance to that of TEDDY with dense weights (scenario 4), across varying pruning simulations, underscoring the robustness of our weight sparsification method. Additional studies to further evaluate this effect on other architectures will be included in the later revision.
>
> ---
>
> **[On Novelty of Multi-hop Consideration]**
>
> **Q3-2.** We acknowledge that multi-hop subgraph analysis is a well-established concept in graph learning. Nevertheless, the application of leveraging **a sole degree information** in multi-level consideration in node classification task has been under-explored. The referenced work [9] provided mainly addresses the **performance enhancement** on graph-level tasks by incorporating a broader range of neighborhood **features** and structural complexities, rather than concentrating on the pruning aspect of GNNs to **preserve the original performance**. Our approach, which specifically considers multi-hop **edge degree** for GNN pruning, introduces a novel angle within this domain.

---

> > ### Comment · Reviewer_k6Ge · 2023-11-22
> >
> > Thank you for your response. Could you provide insights on the isolated impact of the proposed edge sparsification technique? How does the performance of the proposed edge pruning, as outlined in Equation (5), compare to basic high-degree pruning (as shown in Figure 1, Figure 10, and Table 2) when both the distillation loss and weight sparsification are disabled?
> >
> > Based on the observations from the top of Figure 18, it appears that weight sparsity has a minimal impact on performance. Additionally, the effectiveness of the distillation loss has been shown to be quite significant. Considering these factors, it would be interesting to explore the specific performance benefits attributable to the proposed edge sparsification technique. This would help in understanding the extent to which the edge pruning method, as a standalone component, contributes to the overall performance gains.

---

> > > ### Author Response · Authors · 2023-11-23
> > > **Response to Reviewer k6Ge**
> > >
> > > **[Isolated Impact of TEDDY’s Edge Sparsification]**
> > >
> > > Following the reviewer’s insightful suggestions, we present a comparison of a sole graph sparsification performance between multi-level considered pruning approach of TEDDY, basic high-degree pruning, and baselines. We applied uniform three random seeds to all settings, to guarantee the fair comparisons.
> > >
> > > As depicted in the below table, the multi-level considered pruning approach of our method demonstrates superiority over all baselines, including a naive high-degree pruning, particularly on the Pubmed dataset. Thus, the results clearly demonstrate the efficacy of our degree-reflected sparsification technique, as well as the significance of the multi-level consideration. We will provide additional experimental results on GCN and GAT in the later version.
> > >
> > > Again, we are deeply grateful to the reviewer for thoughtful suggestions. Your feedback has been invaluable in enhancing the quality of our work. We hope our overall responses have addressed your concerns.
> > >
> > > ---
> > > Table1. A sole graph sparsification performance (in percentage) of proposed method, high degree-based pruning, and baselines.
> > >
> > > | Simulations | 1-th                | 5-th                | 10-th               | 15-th               | 20-th               |
> > > |-------------|---------------------|---------------------|---------------------|---------------------|---------------------|
> > > | **Cora**    |                     |                     |                     |                     |                     |
> > > | UGS         | 71.83 ± 2.75        | 69.83 ± 2.96        | 68.27 ± 2.18        | 68.53 ± 2.15        | 67.60 ± 2.49        |
> > > | WD-GLT      | 76.83 ± 0.31                   | 75.47 ± 0.05                  | 73.80 ± 0.93                  | 71.43 ± 1.47                  | 70.97 ± 1.23                   |
> > > | **Ours**    | 73.24 ± 0.36        | **77.60 ± 0.24**    | **76.52 ± 0.27**    | **76.48 ± 0.28**    | **75.78 ± 0.19**    |
> > > | High degree | **77.52 ± 0.59**    | 77.54 ± 1.86        | 75.10 ± 1.46        | 73.76 ± 0.81        | 70.90 ± 1.01        |
> > > | **Citeseer**|                     |                     |                     |                     |                     |
> > > | UGS         | 64.80 ± 2.02        | 65.57 ± 3.35        | 65.37 ± 2.08        | 64.47 ± 1.86        | 63.50 ± 1.84        |
> > > | WD-GLT      | 63.93 ± 2.90        | 64.63 ± 2.38        | 62.93 ± 2.90        | 63.60 ± 1.92        | 62.57 ± 1.50        |
> > > | **Ours**    | **69.64 ± 0.48**    | 68.96 ± 0.15        | 68.78 ± 0.20        | **68.75 ± 0.19**    | **68.58 ± 0.17**    |
> > > | High degree | 67.92 ± 1.14        | **68.98 ± 0.62**    | **68.80 ± 0.58**    | 67.08 ± 0.17        | 65.52 ± 1.03        |
> > > | **Pubmed**  |                     |                     |                     |                     |                     |
> > > | UGS         | 72.57 ± 1.93        | 74.70 ± 2.94        | 74.37 ± 2.25        | 73.27 ± 2.01        | 72.43 ± 2.03        |
> > > | WD-GLT      | 76.43 ± 1.22        | 76.33 ± 2.19        | 77.50 ± 0.99    | 75.87 ± 0.60        | 75.47 ± 0.66        |
> > > | **Ours**    | **78.66 ± 0.17**    | **78.44 ± 0.39**    | **78.18 ± 0.35**        | **77.54 ± 0.34**    | **77.14 ± 0.33**    |
> > > | High degree | 78.12 ± 0.43        | 77.36 ± 0.39        | 76.78 ± 1.25        | 74.82 ± 0.58        | 73.00 ± 0.51        |

---

> > > > ### Comment · Reviewer_k6Ge · 2023-11-23
> > > >
> > > > Thank you for addressing most of my technical concerns in your responses. I'd like to update my rating to "5: marginally below the acceptance threshold".
> > > >
> > > > However, I am still concerned about the presentation of the distillation effect in the paper. The experimental results clearly indicate that distillation is a crucial and significant component. Yet, this aspect is not mentioned until P6, Section 5.2. The earlier sections, up to Section 5.2, primarily focus on edge sparsification, without even a mention of 'distillation'. This approach raises serious doubts for me regarding the structure and narrative style of the paper.
> > > >
> > > > As reviewer mYP6 also points out,
> > > > > one still needs to train a dense network on the entire graph first
> > > >
> > > > This is a somewhat significant shortcoming in terms of efficiency. However, it is somewhat hidden from the reader and is not made visible until Page 6 and Section 5.2. To the reviewer, for a paper that emphasizes empirical performance, it's more critical to clearly identify the most effective components in the main text, rather than adopting a storytelling approach. Being straightforward about the key elements contributing to performance should be prioritized over narrative-oriented writing.

---

> > > > > ### Author Response · Authors · 2023-11-23
> > > > > **Response to Reviewer k6Ge**
> > > > >
> > > > > We are deeply grateful to reviewer k6Ge for insightful suggestions and for raising the score of our paper.
> > > > >
> > > > > Following the reviewer's valuable feedback, we will revise our paper to refine our manuscript to improve clarity and precision, particularly concerning the distillation technique and the utilization of a trained model, in subsequent versions.
> > > > >
> > > > > Once again, we sincerely thank the reviewer for the reassessment of our work.

---

> ### Author Response · Authors · 2023-11-21
> **Response to Reviewer k6Ge (4/8)**
>
> **[On Theoretical Claim]**
>
> **Q4.** Note that the generalization error bound for GNN [10] is not our contribution, but we can offer theoretical evidences for our TEDDY based on the results in [10]. In fact, the most challenging component in computing the Lipschitz constant for practical GNN families is the ***non-smooth ReLU function**.* Theoretically, [10] considered a slightly relaxed ReLU function that makes ReLU continuously differentiable. In such cases, since every part of the function becomes smooth, it is possible to compute the Lipschitz constant. It is anticipated that the Lipschitz constant of the original GNN can also be approximated to some extent through this relaxed activation function. Regarding the quantity $\frac{deg_{max}+1}{deg_{min}+1}$, the action of removing low-degree ***edges (not nodes)*** tends to increase the generalization bound by consistently making $deg_{min}$ monotonically decreasing, thereby inducing worse generalization. Similarly, removing high-degree ***edges*** also consistently makes $deg_{max}$  monotonically decreasing. In contrast to the previous case, this tends to reduce the generalization gap, encouraging better generalization.

---

> ### Author Response · Authors · 2023-11-21
> **Response to Reviewer k6Ge (5/8)**
>
> **[On Eq. (5)]**
>
> **Q5.** As we explicitly stated in Section 3.1, TEDDY is specifically tailored for *undirected graphs*, and thus does not directly address directed graphs. Although $T_{edge}$ is symmetric in directed graphs as the reviewer pointed out, directed graphs were out of scope for the design of $T_{edge}$.

---

> ### Author Response · Authors · 2023-11-21
> **Response to Reviewer k6Ge (6/8)**
>
> **[On distillation]**
>
> **Q6.** We thank the reviewer for raising an important point. Our findings, as illustrated in Figure 17 and 18, do indicates an enhancement in performance due to the model distillation. However, our analysis emphasizes the significance of 2-hop degree based edge score, $T_{edge}$, in Figure 18 in revised paper. As demonstrated in the figure, the design of edge scores is more critical, since any configuration with the baseline including distillation fails to achieve better pruning performance than our method with a sole $T_{edge}$ (represented as Ours w/o $\mathcal L_{dt}$).
>
> Moreover, the use of pretrained models is a common initial step in iterative pruning approaches as well, since they fully exploit the dense graph and GNNs at the initial pruning stage. Our method is tailored for practical scenarios in mind: for example, facilitating the deployment of GNNs on local devices with performance equivalent to fully trained models on central servers. Hence, the term **“single training”** in our paper refers to the elimination of the iterative cycle of retraining GNNs for each pruning level. Instead, our TEDDY accomplishes the sparsification in a single training phase when given the sparsity levels and trained GNNs, promoting a significant efficiency, as displayed in Table 3-5 in the revised version.

---

> ### Author Response · Authors · 2023-11-21
> **Response to Reviewer k6Ge (7/8)**
>
> **[On Superiority over Iterative Methods]**
>
> **Q7.** Our experimental results demonstrate superior performance compared to baselines across various graph sizes, underscoring our method's robust generalization ability. This is particularly evident in Section 6.1, where TEDDY, when utilizing GAT as the base architecture, shows a performance enhancement ranging from 12.8% to 20.4% over the optimal performances of baseline models. Additionally, TEDDY consistently exhibits stable performance across different datasets, as highlighted in our inductive semi-supervised node classification experiments. For instance, on the Pubmed dataset, TEDDY maintains decent accuracy despite the significant scarcity in the ratio of training nodes (20% for training vs. 80% for inference). These results confirm TEDDY's outstanding generalization abilities, marking it as a superior choice across diverse datasets compared to conventional iterative baselines.

---

> ### Author Response · Authors · 2023-11-21
> **Response to Reviewer k6Ge (8/8)**
>
> **[On Efficiency of PGD]**
>
> **Q8.** The existing iterative approaches to parameter sparsification involve the full training of an $\ell_1$-regularized mask to identify crucial parameter coordinates. After training, $p_\theta$% of coordinates with the smallest signals in the mask are zeroed out, and the obtained mask is used to train a subnetwork to obtain the graph lottery ticket. Therefore, iterative methods inevitably require two rounds of training to obtain the mask, and achieving the desired target sparsity ratio necessitates multiple times of training process. In contrast, TEDDY can obtain sparse parameters in a single training pass using Projected Gradient Descent on the $\ell_0$-ball for any parameter initialization. Furthermore, since the parameters are already sparse during the training process, TEDDY can also reduce the training time. To validate this, we conduct wall-clock time comparisons for parameter sparsification across each method. Toward this, we use the original dense adjacency matrix while pruning only model parameters for fair comparisons. For your convenience, we attached the below tables in Table 6-8 in the revised paper.
>
> ---
>
> **References**
>
> [1] Provably Powerful Graph Networks, NeurIPS 2019.
>
> [2] Recipe for a general, powerful, scalable graph transformer, NeurIPS 2022.
>
> [3] Specformer: Spectral Graph Neural Networks Meet Transformers, ICLR 2023.
>
> [4] Masked Label Prediction: Unified Message Passing Model for Semi-Supervised Classification, IJCAI 2021.
>
> [5] NAGphormer: A Tokenized Graph Transformer for Node Classification in Large Graphs, ICLR 2023.
>
> [6] Towards Deep Attention in Graph Neural Networks: Problems and Remedies, ICML 2023.
>
> [7] A Unified Lottery Ticket Hypothesis for Graph Neural Networks, ICML 2021.
>
> [8] Rethinking Graph Lottery Tickets: Graph Sparsity Matters, ICLR 2023.
>
> [9] Weisfeiler and Leman Go Neural: Higher-order Graph Neural Networks, AAAI 2019.
>
> [10] Towards Understanding Generalization of Graph Neural Networks, ICML 2023.

---

> > ### Author Response · Authors · 2023-11-21
> > **Response to Reviewer k6Ge (8/8, Experiments on Small- and Medium-scale Datasets)**
> >
> > |   Datasets & GNNs    |    Methods    | 1-st               | 5-th               | 10-th              | 15-th              | 20-th              |
> > |-------|--------|---------------------|---------------------|---------------------|---------------------|---------------------|
> > | **Cora** |        |                     |                     |                     |                     |                     |
> > | GCN   | UGS    | 6.09 ± 0.81        | 5.42 ± 0.22        | 5.71 ± 0.47        | 5.47 ± 0.29        | 5.51 ± 0.22        |
> > |       | WD-GLT | 62.78 ± 3.06       | 60.89 ± 0.96       | 61.38 ± 0.93       | 61.82 ± 1.15       | 61.92 ± 0.87       |
> > |       | TEDDY  | **2.39 ± 0.06**    | **2.43 ± 0.04**    | **2.44 ± 0.10**    | **2.41 ± 0.04**    | **2.41 ± 0.04**    |
> > | GAT   | UGS    | 6.00 ± 1.04        | 5.58 ± 0.34        | 5.60 ± 0.29        | 5.50 ± 0.18        | 5.48 ± 0.19        |
> > |       | WD-GLT | 61.95 ± 3.52       | 61.45 ± 2.13       | 61.40 ± 1.90       | 59.99 ± 1.48       | 60.27 ± 2.29       |
> > |       | TEDDY  | **2.43 ± 0.09**    | **2.47 ± 0.09**    | **2.50 ± 0.15**    | **2.45 ± 0.08**    | **2.44 ± 0.08**    |
> > | GIN   | UGS    | 4.33 ± 0.74        | 4.15 ± 0.46        | 4.05 ± 0.18        | 3.99 ± 0.29        | 3.87 ± 0.20        |
> > |       | WD-GLT | 61.98 ± 3.15       | 60.63 ± 0.39       | 60.38 ± 0.46       | 60.46 ± 0.30       | 60.42 ± 0.45       |
> > |       | TEDDY  | **1.76 ± 0.03**    | **1.80 ± 0.05**    | **1.82 ± 0.04**    | **1.81 ± 0.02**    | **1.81 ± 0.01**    |
> > | **Citeseer** |        |                     |                     |                     |                     |                     |
> > | GCN   | UGS    | 8.84 ± 2.27        | 7.87 ± 1.16        | 7.56 ± 0.81        | 7.56 ± 0.62        | 7.43 ± 0.65        |
> > |       | WD-GLT | 56.97 ± 2.45       | 56.05 ± 0.86       | 55.96 ± 1.33       | 55.38 ± 0.60       | 55.29 ± 0.53       |
> > |       | TEDDY  | **3.27 ± 0.04**    | **3.37 ± 0.06**    | **3.46 ± 0.03**    | **3.50 ± 0.01**    | **3.60 ± 0.05**    |
> > | GAT   | UGS    | 8.42 ± 1.17        | 7.95 ± 0.58        | 7.91 ± 0.70        | 7.89 ± 0.49        | 7.77 ± 0.58        |
> > |       | WD-GLT | 57.67 ± 2.29       | 56.90 ± 0.57       | 56.47 ± 0.80       | 57.30 ± 1.17       | 56.51 ± 0.57       |
> > |       | TEDDY  | **3.35 ± 0.05**    | **3.39 ± 0.04**    | **3.50 ± 0.03**    | **3.55 ± 0.01**    | **3.66 ± 0.05**    |
> > | GIN   | UGS    | 8.28 ± 1.00        | 8.37 ± 1.01        | 8.18 ± 0.84        | 8.16 ± 0.95        | 7.81 ± 0.64        |
> > |       | WD-GLT | 57.50 ± 2.02       | 56.11 ± 0.26       | 56.19 ± 0.54       | 56.88 ± 1.40       | 56.38 ± 0.71       |
> > |       | TEDDY  | **3.42 ± 0.05**    | **3.46 ± 0.04**    | **3.57 ± 0.04**    | **3.62 ± 0.02**    | **3.71 ± 0.05**    |
> > | **Pubmed** |        |                     |                     |                     |                     |                     |
> > | GCN   | UGS    | 9.21 ± 0.96        | 8.48 ± 0.46        | 8.66 ± 0.62        | 8.28 ± 0.20        | 8.38 ± 0.42        |
> > |       | WD-GLT | 202.28 ± 1.98      | 201.13 ± 1.54      | 200.82 ± 2.08      | 201.07 ± 1.53      | 201.16 ± 1.22      |
> > |       | TEDDY  | **3.91 ± 0.08**    | **3.88 ± 0.07**    | **3.90 ± 0.11**    | **3.89 ± 0.10**    | **3.91 ± 0.12**    |
> > | GAT   | UGS    | 13.69 ± 0.97       | 13.18 ± 0.23       | 13.05 ± 0.24       | 13.07 ± 0.21       | 13.12 ± 0.14       |
> > |       | WD-GLT | 66.03 ± 0.75       | 65.68 ± 0.06       | 65.65 ± 0.12       | 65.66 ± 0.14       | 65.62 ± 0.09       |
> > |       | TEDDY  | **6.33 ± 0.05**    | **6.32 ± 0.03**    | **6.33 ± 0.04**    | **6.32 ± 0.04**    | **6.33 ± 0.02**    |
> > | GIN   | UGS    | 8.68 ± 0.78        | 8.38 ± 0.15        | 8.37 ± 0.19        | 8.35 ± 0.11        | 8.26 ± 0.07        |
> > |       | WD-GLT | 157.18 ± 0.48      | 157.03 ± 1.12      | 156.43 ± 0.93      | 157.75 ± 0.59      | 157.88 ± 0.51      |
> > |       | TEDDY  | **4.07 ± 0.04**    | **4.04 ± 0.03**    | **4.06 ± 0.01**    | **4.04 ± 0.04**    | **4.04 ± 0.06**    |

---

> > > ### Author Response · Authors · 2023-11-21
> > > **Response to Reviewer k6Ge (8/8, Experiments on Large-scale Datasets)**
> > >
> > > |    Datasets & GNNs      |  Methods     | 1-st               | 5-th               | 10-th              | 15-th              | 20-th              |
> > > |----------|------|---------------------|---------------------|---------------------|---------------------|---------------------|
> > > | **Arxiv**  |      |                     |                     |                     |                     |                     |
> > > | GCN      | UGS  | 130.66 ± 15.32      | 125.78 ± 13.97      | 128.90 ± 14.12      | 113.48 ± 11.06      | 119.41 ± 11.84      |
> > > |          | WD-GLT | N/A               | N/A                 | N/A                 | N/A                 | N/A                 |
> > > |          | TEDDY | **62.68 ± 1.29**   | **57.10 ± 1.33**    | **51.75 ± 0.97**    | **47.55 ± 1.21**    | **45.07 ± 1.20**    |
> > > | SAGE     | UGS  | 111.71 ± 7.45       | 104.29 ± 9.21       | 106.31 ± 8.13       | 94.02 ± 7.78        | 90.71 ± 6.59        |
> > > |          | WD-GLT | N/A               | N/A                 | N/A                 | N/A                 | N/A                 |
> > > |          | TEDDY | **64.03 ± 0.16**   | **64.03 ± 0.09**    | **64.44 ± 0.17**    | **64.13 ± 0.07**    | **64.34 ± 0.13**    |
> > > | **Reddit**  |      |                     |                     |                     |                     |                     |
> > > | GCN      | UGS  | 210.36 ± 51.92      | 200.75 ± 43.13      | 197.71 ± 47.39      | 173.81 ± 39.13      | 170.48 ± 42.65      |
> > > |          | WD-GLT | OOM               | OOM                 | OOM                 | OOM                 | OOM                 |
> > > |          | TEDDY | **70.34 ± 0.05**   | **65.69 ± 0.11**    | **58.43 ± 0.03**    | **51.72 ± 0.15**    | **47.03 ± 0.13**    |
> > > | SAGE     | UGS  | 203.75 ± 29.72      | 194.58 ± 25.91      | 188.02 ± 31.74      | 181.86 ± 35.12      | 182.97 ± 26.94      |
> > > |          | WD-GLT | OOM               | OOM                 | OOM                 | OOM                 | OOM                 |
> > > |          | TEDDY | **80.63 ± 0.11**   | **78.21 ± 0.06**    | **67.93 ± 0.13**    | **63.11 ± 0.04**    | **51.03 ± 0.16**    |

---

> ### Author Response · Authors · 2023-11-23
> **Response to Reviewer k6Ge**
>
> **[Additional Insights on Training Noise]**
>
> In response to the reviewer’s constructive feedback, we reported average pruning performance of our TEDDY and baselines across all sparsities, represented in Figure 13 and 14 in the revised manuscript. To ensure fair comparisons, we adopted uniform random seeds to all methods for each setting. Owing to the constraints of the rebuttal period, we present performance results of NAGphormer [1] and Specformer [2] on Cora and Citeseer datasets, averaged over 3 runs (Note that the eigendecomposition of Pubmed dataset required prohibitive time consumptions). Meanwhile, we adopted uniform five random seeds for UniMP [3].
>
> As depicted in the figures, our TEDDY still accomplishes prominent performance, surpassing all baselines when equipped with UniMP as a backbone. In particular, UGS [4] and WD-GLT [5] exhibit severe unstability as the sparsity increases, whereas the performance of TEDDY remains stable across all simulations with notably small standard deviations. Our method also achieves decent performance in NAGphormer and Specformer, notably in NAGphormer on the Cora dataset, with considerable performance enhancement over the original result. Hence, the results clearly demonstrate our method’s versatility and robustness across diverse foundational architectures.
>
> ---
> **References**
>
> [1] NAGphormer: A Tokenized Graph Transformer for Node Classification in Large Graphs, ICLR 2023.
>
> [2] Specformer: Spectral Graph Neural Networks Meet Transformers, ICLR 2023.
>
> [3] Masked Label Prediction: Unified Message Passing Model for Semi-Supervised Classification, IJCAI 2021.
>
> [4] A Unified Lottery Ticket Hypothesis for Graph Neural Networks, ICML 2021.
>
> [5] Rethinking Graph Lottery Tickets: Graph Sparsity Matters, ICLR 2023.

---

### Author Response · Authors · 2023-11-21
**General Response**

Dear reviewers and AC,

We sincerely appreciate your dedicated time and insightful feedback on our manuscript.

As highlighted by the reviewers, we introduce TEDDY, a novel edge sparsification framework in a one-shot approach (vxbf, mYP6, SQ4D, QTmY). This has been appreciated for its simplicity (vxbf, SQ4D), conciseness (k6Ge), and novelty (mYP6). Our method is inspired by a critical observation (SQ4D, QTmY) on the importance of preserving low-degree edges, which we have substantiated both empirically and theoretically (vxbf, mYP6). TEDDY has demonstrated its ability to outperform existing state-of-the-art iterative graph pruning methods (k6Ge, vxbf, SQ4D), effectively balancing efficiency and effectiveness (mYP6, QTmY).

In accordance with your constructive comments, we have carefully made the revision with the following additional discussions and experiments:

- Elucidation of the novel aspects distinguishing TEDDY from contemporary sparsification techniques.
- A comparison of pruning durations between TEDDY and other existing approaches.
- An extensive ablation study, covering various scenarios including:
    - TEDDY with all components integrated.
    - TEDDY without the distillation loss.
    - TEDDY employing magnitude-based pruning.
    - TEDDY with dense weights.
    - Baseline with all components integrated.
    - Baseline with the distillation loss.
    - Baseline employing $\ell_0$-based projected gradient descent.

We have marked the revisions in blue for your convenience. We believe that these adjustments better deliver the effectiveness of TEDDY, offering a valuable contribution to the ICLR community.

Thank you once again for your guidance and support.

Authors.

---

### Author Response · Authors · 2023-11-22
**Gentle Reminder**

Dear Reviewers and Area Chair,

We kindly request that you review our responses and the revision at your earliest convenience, since this week (22nd ~ 23rd) marks the end of our interaction period.

We have carefully responded to your feedbacks and faithfully reflected them in the rebuttal with the revised paper, as well as considerable additional experimental results including comprehensive ablation studies.

We sincerely thank you for your time and efforts in reviewing our paper, and your insightful and constructive comments.

Thank you,

Authors

---

### Author Response · Authors · 2023-11-23
**Gentle Reminder for Final Hours of Discussion**

Dear Reviewers and Area Chair,

As the rebuttal period is swiftly approaching its conclusion, we kindly remind you to review our responses and revisions at your earliest convenience. Your attention to our revisions and responses will be profoundly valuable. We deeply appreciate the time and effort you have dedicated to evaluating our work.

Sincerely,

Authors

---

### Meta-Review · Area_Chair_TQJm · 2023-12-05

**Metareview:**

The paper introduces a new method for graph sparsification to speed-up learning. The graph sparsification method is based on three main techniques: degree-based graph sparsification, distillation by matching the classification logits of the model pre-trained on the whole graphs; parameter sparsification by using a subset of parameters.

The paper studies an important problem and introduces a new simple method with good experimental results. Overall, the paper is an interesting contribution and it would be a good addition to the ICLR program.

Nevertheless, the committee would strongly suggest few changes to the presentation before publication:
- the paper should emphasize more the role of distillation that seems central in the experimental results.
- the paper should include a discussion on the complexity of the method in the final version

**Justification For Why Not Higher Score:**

The paper is interesting but it has few shortcoming listed above.

**Justification For Why Not Lower Score:**

The paper presents a new method for an important problem with convincing experimental results

---

### Decision · Program_Chairs · 2024-01-16

Accept (poster)